

# Hourly surface nitrogen dioxide retrieval from GEMS tropospheric vertical column densities: Benefit of using time-contiguous input features for machine learning models

Janek Gödeke[1], Andreas Richter[2], Kezia Lange[2], Peter Maaß[1], Hyunkee Hong[3], Hanlim Lee[4], and Junsung Park[4]

[1]Center for Industrial Mathematics, University of Bremen, Germany
[2]Institute of Environmental Physics, University of Bremen, Germany
[3]National Institute of Environmental Research, Environmental Satellite Center, Korea
[4]Pukyong National University, Korea

**Correspondence:** Janek Gödeke (janek-goedeke@uni-bremen.de)

**Abstract.** Launched in 2020, the Korean Geostationary Environmental Monitoring Spectrometer (GEMS) is the first geostationary satellite mission for observing trace gas concentrations in the Earth's atmosphere. Observations are made over Asia. Geostationary orbits allow for hourly measurements, which leads to a much higher temporal resolution compared to daily measurements taken from low Earth orbits, such as by the TROPOspheric Monitoring Instrument (TROPOMI) or Ozone Monitoring Instrument (OMI). This work estimates the hourly concentration of surface $NO_2$ from GEMS tropospheric $NO_2$ vertical column densities (tropospheric $NO_2$ VCDs) and additional meteorological features, which serve as inputs for Random Forests and linear regression models. With several measurements per day, not only the current observations but also those from previous hours can be used as inputs for the machine learning models. We demonstrate that using these time-contiguous inputs leads to reliable improvements regarding all considered performance measures, such as Pearson correlation or Mean Square Error. For Random Forests, the average performance gains are between $4.5\%$ and $7.5\%$, depending on the performance measure. For linear regression models, average performance gains are between $7\%$ and $15\%$. For performance evaluation, spatial cross validation with surface in-situ measurements is used to measure how well the trained models perform at locations where they have not received any training data. In other words, we inspect the models' ability to generalize to unseen locations. Additionally, we investigate the influence of tropospheric $NO_2$ VCDs on the performance. The region of our study is Korea.

## 1 Introduction

The concentration of nitrogen dioxide ($NO_2$) near the earth's surface is of significant interest for several reasons. On the one hand, $NO_2$ is not only a precursor of the health hazard and air pollutant ozone, but $NO_2$ also has a direct negative impact on human health. On the other hand, it is linked to environmental issues such as acid rain, see e.g. the book of Jacob (2000).

At present, surface $NO_2$ is measured by networks of ground-based in situ monitoring stations. However, due to the limited number of such stations, they cannot provide global information about the surface $NO_2$ concentration. This limitation is one of the reasons why satellite remote sensing has become popular for deriving global estimates of surface $NO_2$. In short, satellites



detect the fingerprint of NO$_2$ within the backscattered solar radiation due to its strong absorption of light in the wavelength range of $(350\text{-}500)$ nm. One of the first studies on deriving surface NO$_2$ from remote sensing observations was conducted by Lamsal et al. (2008) across the USA and Canada. In their study, surface NO$_2$ was estimated by applying the NO2 vertical

distribution derived by chemical transport models to tropospheric NO$_2$ vertical column densities (tropospheric NO$_2$ VCDs), the latter being obtained from measurements of the Ozone Monitoring Instrument (OMI, Levelt et al. (2006)). Numerous further studies followed, also utilizing chemical transport models and observations from satellites in low-earth orbits. For example, we refer to the studies of Lamsal et al. (2010), Lamsal et al. (2013), Bechle et al. (2013), Wang and Chen (2013), Kharol et al. (2015), Geddes et al. (2016), Gu et al. (2017), Cooper et al. (2020) and Cooper et al. (2022). Not only OMI data has been

considered, but also observations from, e.g., the Global Ozone Monitoring Experiment (GOME, Burrows et al. (1999)), the Scanning Imaging Absorption Spectrometer for Atmospheric Chartography (SCIAMACHY, Bovensmann et al. (1999)), or the TROPOspheric Monitoring Instrument (TROPOMI, Veefkind et al. (2012)).

During the last ten years, machine learning approaches received increasing attention in surface NO$_2$ estimation from satellite remote sensing observations. One advantage is the shorter computation time, once the model has been trained. Diverse machine

learning models have been used for this task, exploiting not only tropospheric NO$_2$ VCDs as an input, but also additional input features for improving the model's performance, such as meteorological parameters, traffic density or population information. Studies that considered observations from satellites in low-earth orbits have been conducted for example by Kim et al. (2017), Jiang and Christakos (2018), de Hoogh et al. (2019), Chen et al. (2019), Di et al. (2020), Qin et al. (2020), Kim et al. (2021), Chan et al. (2021), Dou et al. (2021), Ghahremanloo et al. (2021), Li et al. (2022), Wei et al. (2022), Huang et al. (2023),

Shetty et al. (2024). For a detailed review on the methods used, the input features included, the regions of consideration and the achieved performances we refer to the work of Siddique et al. (2024).

Satellites in low-earth orbits such as OMI or TROPOMI pass the same region in mid and low latitudes once a day, which means they can provide at best one measurement per day and location. If the area is cloud-covered during the time of observation, no measurement can be taken, which makes the data coverage even more limited. This is why most studies estimated daily,

monthly or annual surface NO$_2$ concentrations. Nevertheless, it is to be mentioned that there are a few studies that estimated hourly NO$_2$. As an example, Kim et al. (2021) linearly interpolated daily tropospheric NO$_2$ VCDs to an hourly resolution, from which they estimated hourly surface NO$_2$ concentrations over Switzerland and northern Italy.

In contrast, geostationary satellites permanently observe - more or less - the same region, leading to more data points for a given location that can be used for a prediction algorithm of surface NO$_2$. In particular, these larger datasets make machine

learning approaches even more attractive. The first geostationary satellite instrument for observing trace gas concentrations in the Earth's atmosphere is the Geostationary Environment Monitoring Spectrometer (GEMS, Kim et al. (2020)), which was launched in February 2020 by the Republic of Korea. It provides hourly measurements of radiances over around 20 countries in Asia, among which is Korea. Alongside GEMS, there exists only one more geostationary satellite for monitoring trace gases, namely NASA's TEMPO, which was launched recently in April 2023 and is observing North America. A third geostationary

satellite, ESA's Sentinel-4 mission, is foreseen for launch in 2025 and will monitor Europe.




Until now, only a few studies have been done about hourly surface $NO_2$ retrieval from geostationary observations: Zhang et al. (2023) presented a scientific GEMS $NO_2$ product (POMINO-GEMS), which empirically corrects for overestimation and stripe artifacts in the operational GEMS $NO_2$ product. They then converted their tropospheric $NO_2$ VCDs of 2021 over China to hourly surface $NO_2$ using a chemical transport model. Further studies have been conducted over China exploiting machine
learning approaches. Yang et al. (2023) used a Random Forest regressor for predicting hourly surface $NO_2$ over China from GEMS radiance data at six wavelengths from the UV and visible bands, and some additional meteorological, temporal and spatial features. Furthermore, a multi output Random Forest was used to simultaneously predict five further air pollutants, such as ozone. Although prediction accuracy achieved by the multi output model was slightly worse regarding surface $NO_2$, the overall training time for predicting all six pollutant concentrations was smaller. Ahmad et al. (2024) combined two machine
learning models. First, a Random Forest was used to predict the $NO_2$ mixing heights from meteorological input features. These were then fed into an Extreme Gradient Boosting regressor, together with tropospheric $NO_2$ VCDs from GEMS, temporal and meteorological variables. The study demonstrates the benefit of using the $NO_2$ mixing height as an input.

Hourly surface $NO_2$ has also been predicted from GEMS observations over Korea, the region considered in this study. Namely within the work of Lee et al. (2024) for the whole year 2022. Therein, total, instead of tropospheric $NO_2$ VCDs were
used as the only input of a (linear) Mixed Effect Model to predict surface $NO_2$. Their model is a piece-wise defined function, whose output depends not only on the total column of $NO_2$, but also on the day and hour as well as the region at which the prediction is to be made. For that, Korea was divided into nine regions, which presumably leads to a region-wise more direct relation between surface $NO_2$ and column densities of $NO_2$. In other words, implicitly, spatial and detailed temporal information are also exploited in their approach. This makes their model specialized to Korea and the year 2022.

Another work that predicted surface $NO_2$ over Korea has been conducted by Tang et al. (2024). Therein, daily surface $NO_2$ concentrations were predicted, instead of hourly surface $NO_2$. Further, they do not use $NO_2$ column densities as an input for a machine learning model. Instead, they inspected the influence of aerosol optical depth, which is part of the GEMS data products. Aerosol optical depth, together with surface $NO_2$ predictions from a chemical transport model and other features such as meteorological parameters, served as inputs for a Random Forest to estimate surface $NO_2$.

In order to train and evaluate machine learning models of surface $NO_2$, in-situ $NO_2$ observations from ground-based networks are used. Within the literature, there are two frequently used strategies for evaluating the performance of a machine learning model for predicting surface $NO_2$. First, standard k-fold cross validation is considered, see for example the works of Ghahremanloo et al. (2021), Chan et al. (2021), Yang et al. (2023), Ahmad et al. (2024). This means that the whole dataset is randomly split into $k$ equally sized subsets. One of them serves as the test set, whereas the other $k-1$ are used for training the
model. Training and testing is repeated $k$ times, until each subset has served once as a test set. The average test performance (e.g., Pearson correlation) is calculated and represents the final evaluation of the model. For standard k-fold cross validation, data from all available in situ stations is contained in both the training and test datasets (with large probability). However, what if the trained model should afterwards predict surface $NO_2$ at some new location which has not contributed data to the training set? With the result from standard cross validation, it would be impossible to say how reliable the model can generalize to
this unseen location. It may have over-fitted to the locations that it has dealt with during training. Therefore, if global charts





covering large areas like the entirety of Korea are desired, it would be more appropriate to evaluate the model's performance via so-called *spatial k-fold cross validation*. This means the set of available in situ stations is divided into training and test stations, the model gets trained with data from training stations only, and finally its performance in predicting surface NO$_2$ at the test stations is evaluated. Unsurprisingly, performance measured with spatial cross validation is indeed worse compared to

standard cross validation, which has been observed, e.g., within the studies of Ghahremanloo et al. (2021), Chan et al. (2021), Yang et al. (2023), Tang et al. (2024). In our work we will focus on spatial k-fold cross validation, as we wish to inspect how well a model can generalize to unseen locations.

### 1.1 Goals of this study

Due to the hourly measurements GEMS provides over the same region, it is natural to ask whether one can directly benefit
from the time resolution itself and not only from the resulting larger size of the dataset. Hence, we propose to train a machine learning model $\varphi$ that predicts surface NO$_2$ at some location $z$ and time $t$ not only from corresponding tropospheric NO$_2$ VCD and meteorological data at time $t$, but also gets these inputs at $(k-1) \in \mathbb{N}_0$ previous hours ($\mathbb{N}_0$ are the natural numbers including zero). This means the model is a mapping $\varphi : \mathbb{R}^{pk} \to \mathbb{R}$, where $p$ is the number of different features:

$$
\text{input}(z,t) := \begin{pmatrix} \text{tropospheric NO}_2 \text{ VCD}(z,t) \\ \vdots \\ \text{tropospheric NO}_2 \text{ VCD}(z,t-k+1) \\ \text{meteorological features}(z,t) \\ \vdots \\ \text{meteorological features}(z,t-k+1) \end{pmatrix} \longmapsto \varphi(\text{input}(z,t)) \approx \text{surface NO}_2(z,t)
$$

Here $t-j$ refers to the the time $j$ hours before $t$. In all what follows $k$ is also called as *time-contiguity* of the input features, as it determines at how many times each input feature is included in the whole input vector. Note that $k=1$ stands for the case that only input features at current time $t$ are included. Of course, one could also use features at later times $t+j$. However, we did not observe additional benefit from including features at times $t+1$ and $t+2$. This is why, for simplicity and better readability, we focus on making predictions based on previous-time features in this work.

Our main aim is to inspect whether by using inputs with higher time-contiguity $k$, the performance of the model in predicting surface NO$_2$ at unseen locations will increase. Unseen locations are locations from which the model has not seen any training data. As it will turn out, it is indeed beneficial to use larger time-contiguity $k > 1$ for the machine learning models of our consideration, namely Random Forests and linear regressors. To the best of our knowledge, this observation has not been made in the literature, yet. Regarding work on non-geostationary satellite data, the usage of time-contiguous tropospheric NO$_2$
VCDs is simply impossible, as only single measurements per day are available. We further carefully design experiments that are suitable to answer our main research question about the benefit of time-contiguous inputs. Last but not least, we inspect the influence of tropospheric NO$_2$ VCDs on the models' ability to predict surface NO$_2$ as well as its influence on the benefit





from time-contiguous inputs. This is of interest as it addresses the question of how useful and necessary satellite observations of $NO_2$ are for the prediction of surface $NO_2$ concentrations.

## 1.2 Outline

In Section 2 we describe the different sources of data included in our study. Furthermore, we describe the construction of the datasets used for training machine learning models in our study and give a mathematical description for these datasets. Afterwards, we describe in Section 3.2 the experiments that provide a clear insight into the research questions, e.g. whether time-contiguous inputs can enhance the quality of surface $NO_2$ predictions. We also discuss different loss functions for measuring the performance of trained models on the test dataset. Section 4 serves as a quick recap of the machine learning models used in this study. Finally, we present and discuss the results of our experiments in Section 5.

## 2 Data

In our study, we exploit two data sources for the prediction of surface $NO_2$. The first source are tropospheric $NO_2$ VCDs derived from GEMS measurements, and the second is meteorological data from the ERA5 dataset (Hersbach et al. (2018)). Further, measurements of surface $NO_2$ at in situ stations from the air quality network of Korea serve as the ground truth in this study. This section begins with a brief description of these data sources, followed by a description of the data pre-processing steps. In particular, we explain how the VCDs were paired with ERA5 and in situ data, and how time-contiguous datasets were constructed. For clarity, we provide mathematical definitions of these time-contiguous datasets.

### 2.1 Data sources

#### 2.1.1 GEMS tropospheric $NO_2$ vertical column densities

GEMS is a UV-visible imaging spectrometer onboard the geostationary satellite GK2B. At its launch on 18 February 2020, GEMS was the first geostationary air quality monitoring mission. GEMS is located over the Equator at a longitude of 128.2°E and covers a large part of Asia (5°S-45°N and 75°E-145°E) on an hourly basis. With four different scan modes, which all include Korea, the field of regard (FOR) shifts westward with the Sun. During daytime, GEMS provides up to ten observations according to the season and location with a spatial resolution at Seoul of 3.5 km × 8 km. The GEMS irradiance and radiance measurements in the UV-visible spectral range can be used to derive column amounts of, for example, ozone ($O_3$), sulfur dioxide ($SO_2$), and $NO_2$, but also cloud and aerosol information (Kim et al. (2020)). For this study, we use the tropospheric $NO_2$ VCD product.

During the time of this study, the operational GEMS L2 tropospheric $NO_2$ VCD product was available in v2. This version was evaluated by, e.g., Oak et al. (2024) and Lange et al. (2024), showing that it is high biased compared to the TROPOMI tropospheric $NO_2$ VCD product and ground-based tropospheric $NO_2$ VCD data sets. Additionally, the v2 product showed enhanced scatter. As preparation for the European geostationary instrument on Sentinel-4, the Institute of Environmental Physics





at the University of Bremen (IUP-UB) has developed a scientific GEMS NO$_2$ product. The GEMS IUP-UB tropospheric NO$_2$ VCD v1.0 product was evaluated by Lange et al. (2024) showing good agreement with the operational TROPOMI NO$_2$ data

and ground-based observations. Here, an earlier version (V0.9) of the same data product was used. Briefly, the retrieval is based on a Differential Optical Absorption Spectroscopy fit in the spectral window 405 – 485 nm, using daily GEMS irradiances as background spectra. The stratospheric correction is based on a variant of the STREAM algorithm of Beirle et al. (2016) and tropospheric vertical columns are computed using airmass factors applying the tropospheric NO$_2$ profiles from the TM5 model run performed for the operational TROPOMI product (Williams et al. (2017)). Cloud screening is based on the operational

GEMS cloud product v2 and a threshold of 50% cloud radiance fraction, but no additional cloud correction is performed. Each pixel has a quality indicator (qa value) based on fitting residuals, cloud fraction and surface properties. Here, only data with the highest qa value (good fits, cloud radiance fraction below 50%, no snow or ice detected) are used.

### 2.1.2 Meteorological data

In order to predict surface NO$_2$, it would not be sufficient to use tropospheric NO$_2$ VCDs as the only source of information.

This is because VCDs represent integrals over the entire troposphere, capturing contributions from NO$_2$ at various altitudes, not just near the surface. A common strategy is to incorporate additional meteorological features into the prediction of surface NO$_2$, see for example the works of Di et al. (2020), Qin et al. (2020), Ghahremanloo et al. (2021), Chan et al. (2021), Li et al. (2022), Yang et al. (2023). In our study, we utilize meteorological features from the ERA5 data set, the fifth generation reanalysis by the European Centre for Medium-Range Weather Forecasts (ECMWF), which provides comprehensive global

climate and weather data for the past eight decades (Hersbach et al. (2018)).

Our selection of meteorological features is partially inspired by the choices made in the aforementioned studies, including variables such as boundary layer height, wind components, surface temperature or pressure. The 18 features from ERA5 that are considered during this study are listed in Table B1, where we use the same nomenclature as in the description of the ERA5 dataset, see again Hersbach et al. (2018). In the geographical reference system, the resolution of all meteorological features

is $0.25° \times 0.25°$, which corresponds to approximately $28\,km \times 22\,km$ over Korea. Consequently, ERA5 data is approximately eight times coarser in latitude and three times coarser in longitude than the GEMS tropospheric NO$_2$ VCDs.

### 2.1.3 In situ measurements of surface NO$_2$

In this study, we use in situ surface NO$_2$ measurements from the air quality network AirKorea as the ground truth, provided by the Korean Ministry Of Environment. It can be downloaded from their website https://www.airkorea.or.kr/eng/. There is a

large number of in situ stations in Korea that, among other air pollution-related species, measure surface NO$_2$. We have used data from 637 stations, which are depicted in Fig. 1 (a). Our in situ dataset includes measurements from January 2021 until end of November 2022.





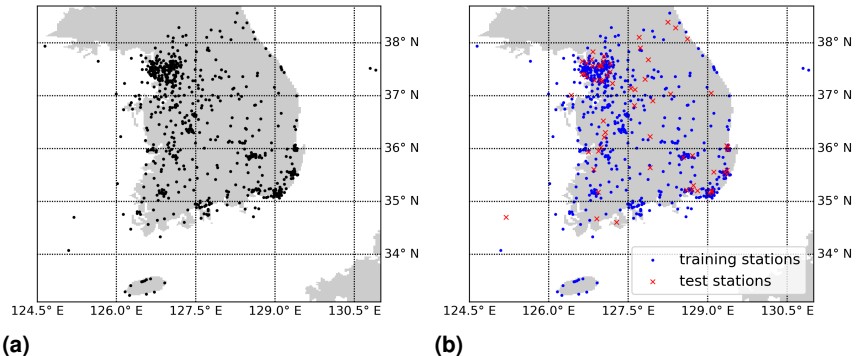

**Figure 1. (a)** Map with the 637 in situ stations from the air quality network of Korea used in this study. **(b)** An exemplary split into 90 % training stations and 10 % test stations, considered during multiple 10-fold spatial cross validation.

## 2.2 Pairing of data sources and data pre-processing

In the following we explain the spatial and temporal pairing of the data sources. Tropospheric $NO_2$ VCDs and meteorological
data possess spatial resolutions, as described in the previous section. Consequently, each data point covers an area (pixel) on the earth's surface, rather than a single point. Here, we associated the location of an in situ station with the VCD pixel or meteorological pixel whose center is nearest to the station's location (longitude, latitude).

In situ measurements of surface $NO_2$ are available as hourly averages. We assume that the averages are taken over the hour following the timestamp given in the file, e.g. from 01:00 UTC to 02:00 UTC for the 01:00 UTC value. On the other hand,
tropospheric $NO_2$ VCDs are based on GEMS observations that have been collected within 30 minutes starting at quarter to the respective hour, e.g. from 01:45 UTC to 02:15 UTC. Most meteorological features are given on the hour, which means at a specific point in time. There is one exception, namely evaporation, which is available as an hourly average starting on the hour, similar to in situ measurements. Since the averages of these data sources are taken over different periods of time, there is not a unique way to pair them temporally. Our approach is the following:

Due to the hourly resolution of all data sources, time $t$ is expressed by $t =$'YYYY/MM/DD/HH' throughout this work. For example, $t =$'2021/01/23/01' refers to 23 January 2021 at 01:00 UTC. We associate with $t$ those in situ measurements of surface $NO_2$ that have started at time $t$ and went on for one hour. In the example, time $t =$'2021/01/23/01' refers to surface $NO_2$ that has been averaged from 01:00 UTC till 02:00 UTC. Regarding tropospheric $NO_2$ VCDs, the same $t$ refers to measurements that have started 45 minutes later. Hence, $t =$'2021/01/23/01' describes the VCDs at some time between 01:45 UTC and 02:15
UTC. Finally, for those meteorological features that are instantaneously on the hour, $t$ stands for the feature's value one hour later at $t + 1$. Thereby, it is closest to the corresponding VCD time frame and further at the end of the corresponding time frame of in situ observations. For example, $t =$'2021/01/23/01' is associated with the meteorological feature at 02:00 UTC. Alternatively, one could have associated the meteorological features one hour earlier, since the measuring time would still be





within the time of the in situ observation. However, in our case, this is going to be included in the time-contiguous datasets
anyway, see the next section.

To sum up, given a location $z$ of an in situ station and some time $t =$'YYYY/MM/DD/HH', we have specified a single data
point $\left( f(z,t), s(z,t) \right)$ that stores surface NO$_2$ $s(z,t)$ combined with the vector of input features $f(z,t)$, which consists of
tropospheric NO$_2$ VCD and meteorological features. As a data pre-processing step we exclude data points that violate any of
the following conditions:

1) All features are available at location $z$ and time $t$ (tropospheric NO$_2$ VCDs and surface NO$_2$ might be missing for some
$z,t$, for example, due to clouds).

2) Tropospheric NO$_2$ VCDs are non-negative. Negative VCDs can occur as the result of measurement noise in the satellite
data or uncertainties in the stratospheric correction. While they are important to use when computing averages in order
to avoid biases, they are not meaningful input for the machine learning model.

3) GEMS qa-value is equal to 1.

Data points $\left( f(z,t), s(z,t) \right)$ that fulfill these conditions are collected within the so-called *data basis*. A data point in the data
basis is not time-contiguous, as it only provides information at a single time $t$ and not at previous hours. The construction of
time-contiguous datasets is described in the next section.

## 2.3 Description of time-contiguous datasets

In the introduction we have motivated the use of time-contiguous inputs for machine learning models in order to predict surface
NO$_2$. For better clarity, we settle down some notation and definitions in a mathematical way.

**Spatial and temporal coordinates:** $Z$ is the set of positions (longitude, latitude) on the earth's surface in terms of
longitude and latitude. Hence, it can be seen as the cartesian product $[-180, 180) \times [-90, 90)$. In this study, we are
dealing with in situ stations in Korea which are located within $[124, 131) \times [33, 39)$, see Fig. 1 (a). These stations will
simply be identified with their location $z \in Z$ in what follows.

$T$ is the set of all measuring times 'YYYY/MM/DD/HH' between January 2021 and November 2022. For example,
'2021/01/23/01' refers to 23 of January 2021 at 01:00 UTC. Note that for given $t \in T$ the expression $t - j$ for $j \in \mathbb{N}$
stands for the time $j$ hours before $t$. For example, for $t =$'2021/01/23/01' and $j = 3$ it is $t - j =$'2021/01/22/22'.

**Surface NO$_2$ and input features:** We recall from the previous section that surface NO$_2$ measured at time $t \in T$ and at
in situ station $z \in Z$ is denoted by $s(z,t)$. As already mentioned, surface NO$_2$ is to be predicted from the tropospheric
NO$_2$ VCD and meteorological variables such as the boundary layer height. These input features at $z \in Z$ and $t \in T$ are
denoted by $f_1(z,t),...,f_p(z,t)$, where $p \in \mathbb{N}$ is the number of considered features (determined by some feature selection
procedure, see Sect. 3.1). At this point, it is only important that $f_1$ denotes the VCDs. For simplicity, we just write
$f(z,t) \in \mathbb{R}^p$ for the vector of all features at location $z$ and time $t$.



**Data pre-processing:** We review the data pre-processing described in the previous section in the light of the mathematical notation. A measurement $f_1(z,t)$ of tropospheric $NO_2$ VCD is called *valid* if it exists (measurements may be missing at some times $t \in T$), if $f_1(z,t) \geq 0$ and if further the GEMS qa-value is equal to 1. For all other features $f_2(z,t),...,f_p(z,t)$ as well as surface $NO_2$ $s(z,t)$ it suffices that the measurement exists in order to be called valid.

In the following we collect all locations and times $(z,t)$ at which we have access to valid measurements. Namely, the

*domain of valid measurements* $\Omega$ is defined as

$$\Omega = \big\{(z,t) \in Z \times T : \text{ and } s(z,t), f_1(z,t),...,f_p(z,t) \text{ are valid}\big\}. \tag{1}$$

**Time-contiguous datasets:** In order to consider time-contiguous measurements, we define for $N \in \mathbb{N}$ the set

$$\Omega_N = \big\{(z,t) \in \Omega : (z,t-j) \in \Omega \text{ for } j = 1,...,N-1\big\}. \tag{2}$$

In other words, $\Omega_N$ collects locations and times $(z,t)$ at which valid measurements do also exist for at least $N-1$

previous hours. Note that $\Omega_N \subseteq \Omega_{N-1} \subseteq \Omega$ for all $N \in \mathbb{N}$ and $\Omega_1$ coincides with $\Omega$, the domain of valid measurements. Given $(z,t) \in \Omega_N$ and $k \in \{1,...,N\}$, this definition allows for building a valid time-contiguous feature vector

$$\begin{pmatrix} f(z,t) \\ f(z,t-1) \\ \vdots \\ f(z,t-k+1) \end{pmatrix} \in \mathbb{R}^{pk}, \tag{3}$$

which can serve as an input for a machine learning model $\varphi_\theta : \mathbb{R}^{pk} \to \mathbb{R}$ to predict surface $NO_2$ $s(z,t)$.

Hence, $\Omega_N$ parameterizes the datasets occurring in our study. In fact, $\Omega_N$ parameterizes $N$ different datasets of feature

vectors paired with surface $NO_2$. They only differ within the time-contiguity $k \in \{1,...,N\}$ of the feature vectors, so how many previous hours (namely $k-1$) shall be considered for each feature (at most $N-1$). Mathematically, these $N$ datasets can be understood as functions $D_{N,k} : \Omega_N \to \mathbb{R}^{pk} \times \mathbb{R}$ mapping $(z,t) \in \Omega_N$ to the feature vector in Eq. (3) paired with surface $NO_2$ at location $z$ and measuring time $t$. Further, $D_{1,1}$ just describes the *data basis* mentioned in the previous section.

The number of elements in $\Omega_N$ - so the size of all datasets $D_{N,k}$ - are listed in Table 1 for $N = 1,...,5$. Hence, if a model is to be trained with time-contiguous inputs ($k > 1$), this comes along with the price of a smaller number of data points. It is to be mentioned that among all features described in the previous section, ERA5 *soil type* and *high vegetation cover* are the only features that do not depend on time $t$. This is why in practice, we never included them $k$ times but rather a single time only, when building the time-contiguous feature vector in Eq. (3) at $(z,t)$. However, for the sake of simplicity,

we neglect this fact within the notation.





**Table 1.** Size of time-contiguous datasets $D_{N,k}$. Note that the size is independent of the time-contiguity $k$. The overall considered time-period covers January 2021 until November 2022.

| N | 1 | 2 | 3 | 4 | 5 |
|---|---|---|---|---|---|
| Number of datapoints | 1,341,642 | 959,458 | 699,777 | 505,719 | 356,117 |

## 3 Experimental setup

In Sect. 3.2, we describe and discuss experiments to inspect our main research questions. Before that, we explain how features were selected for these experiments. Afterwards, we discuss different performance measures and loss functions used to evaluate the quality of the models' prediction of surface $NO_2$ on test data points.

### 3.1 Feature selection

In this study, we considered 23 different features from which we selected 17 for building the feature vectors Eq. (3) used as inputs for the machine learning models. The selected and excluded features are listed in Table B1 and are used in Experiment 1 and Experiment 2, see Sect. 3.2. For the feature selection we proceeded as follows: On the data basis $D_{1,1}$, we considered 200 different splits into $90\%$ training and $10\%$ test stations. For the training data of each split we calculated the Pearson correlation (see Sect. 3.3 for a definition) between in situ measurements of surface $NO_2$ and the respective feature. We selected those features which had an absolute mean correlation larger than $0.1$. It is worth mentioning that in fact for all of the aforementioned 17 features the correlation was larger than $0.1$ in $98\%$ of the splits, whereas this was never the case for the remaining six features. More complex feature selection strategies could be applied in the future. However, during this study we focus on the benefit of time-contiguous inputs and not on the optimal choice of input features.

### 3.2 Experiments

Recall from Sect. 2.3 that $\Omega_N$ is the set of locations and measuring times $(z,t)$ at which all measurements are also available at $(N-1)$ previous hours. Note that $\Omega_N$ does not parameterize a single dataset, but $N$ different datasets $D_{N,k} : \Omega_N \to \mathbb{R}^{pk} \times \mathbb{R}$ via

$$D_{N,k} : (z,t) \longmapsto \left( \begin{pmatrix} f(z,t) \\ f(z,t-1) \\ \vdots \\ f(z,t-k+1) \end{pmatrix}, s(z,t) \right),$$

that only differ in the time-contiguity $k \in \{1,2,...,N\}$ of the time-contiguous feature vector $\big(f(z,t),...,f(z,t-k+1)\big)^T$, defined in Eq. (3).





As mentioned in the introduction, we wish to inspect how well a machine learning model is able to make predictions of surface $NO_2$ at locations from which it has not seen training data. This is why we use multiple (six times) 10-fold spatial cross validation in all experiments. This involves splitting the dataset 60 times randomly into $90\%$ training and $10\%$ test data based on the locations of the in situ stations, see Fig. 1 (b) for a visualization of a single split. Performance is measured on all different test data sets and averaged. Due to the limited number of available in situ stations, significant variance in the model's performance is expected across different splits. Therefore, multiple 10-fold spatial cross validation provides a more reliable estimate for the model's performance, compared to single 10-fold spatial cross validation. In all what follows, whenever it is mentioned that a machine learning model is trained or tested on $D_{N,k}$, it implies that the model is trained or tested solely on those data points in $D_{N,k}$ corresponding to the designated training or test stations. Note that for fixed $N$, surface $NO_2$ that is to be predicted in $D_{N,k}$ is exactly the same for all different $k$. Furthermore, for all models the same 60 splits into training and test stations are considered for spatial cross validation, which ensures perfect comparability.

Let us recall from Sect. 1.1 that our main research question is whether time-contiguous inputs for machine learning models enable higher accuracy for predicting surface $NO_2$. We propose two experiments to gain insight into this question.

**Experiment 1:** Do time-contiguous input features provide additional information?

For fixed $N$ consider the datasets $D_{N,k}$ for different time-contiguities $k = 1, ..., N$. The chosen machine learning model, such as a Random Forest regressor, is trained and tested on $D_{N,k}$ for all 60 splits from spatial cross validation. A comparison is made with respect to different $k$. Fixing $N$ ensures that, regardless of $k$, the same ground truth (surface $NO_2$) is predicted for computing the cross validation scores on the test sets. Additionally, all models are trained with the same number of training data points, eliminating any advantage or disadvantage due to differing dataset sizes. Thus, this experiment provides pure insight into the information gain provided by time-contiguous inputs. We conduct this experiment for all $N \in \{2, 3, 4, 5\}$.

**Experiment 2:** Are time-contiguous input features beneficial in spite of a smaller available dataset?

In the first experiment, the models were trained on the same amount of training data, with the time-contiguity $k$ being the only variable. However, for smaller $k$ there is much more data available that can be used for training the respective models, see Table 1. Therefore, we need to extend the first experiment as follows: We still test performance on $D_{N,k}$ for a fixed $N$. But for different $k$, we train models on $D_{M,k}$ for all $M \in \{k, k+1, ..., 5\}$, so with different amount of training data. Note that in Experiment 1, $M$ has always been set to $N$. These additional investigations are crucial to evaluate whether time-contiguous inputs are beneficial for predicting surface $NO_2$. Even if time-contiguous inputs provide additional information (as seen in the first experiment), why should one use them if training with less or even no time-contiguity on larger datasets yielded better results? Again, we conduct this experiment for all $N \in \{2, 3, 4, 5\}$, where $N$ determines the test datasets.

In a third experiment we analyze the influence of some features to the performance of the machine learning models. Since testing all different combinations of input features for all 15 different training and test cases in Experiment 2 would be out




of scope for this study, we focus on the influence of the tropospheric $NO_2$ VCDs, surface height and latitude, only. Note that longitude has not been included during feature selection due to a low correlation with surface $NO_2$. Tropospheric $NO_2$ VCDs are considered since they represent the feature which shows, among all considered input features, the by far best Pearson correlation to surface measurements of $NO_2$, namely around $0.626$. Note, that the second best correlation is achieved by the boundary layer height and is around $-0.318$, see also Table B1. Regarding the coordinates, there is the risk of spatial overfitting, which would make it more difficult to predict surface $NO_2$ outside of Korea with the same model. Therefore, we check whether the models perform equally well over Korea without having these coordinates as an input.


**Experiment 3:** What is the influence of tropospheric $NO_2$ VCDs, latitude and surface height to the performance?

We compare four different settings of input features:

**Setting 1:** All features selected in Sect. 3.1 are included, which is exactly the setup for Experiments 1 and 2.

**Setting 2:** VCDs are excluded as an input feature.

**Setting 3:** Latitude and surface height are excluded.

**Setting 4:** VCDs, latitude and surface height are excluded.

We also conduct Experiment 2 for Settings 2, 3, and 4, and draw a comparison between these settings regarding different performance measures. Further, within these four settings we inspect the models' ability and reliability of making performance gains when including time-contiguous input features.


### 3.3 Performance measures

Throughout this section, $x^\dagger \in \mathbb{R}^n$ is a vector consisting of $n$ in situ observations of surface $NO_2$, where each coefficient $x_i^\dagger(t_i, z_i) = s(t_i, z_i)$ corresponds to a measurement that has been taken at some time $t_i$ and location (longitude, latitude) $z_i$ of some in situ station. For the sake of simpler notation, we just write $x_i^\dagger$, neglecting the dependence on $t_i$ and $z_i$ within the notation. Similarly, $x \in \mathbb{R}^n$ denotes the predictions for $x^\dagger$ made by some machine learning model, such as linear regression

or Random Forests. In the following, we discuss different performance measures that quantify the gap between the model's prediction $x$ for $x^\dagger$, the observed surface concentration of $NO_2$.

As pointed out in the introduction, spatial cross validation is considered within this research, i.e. data is split into training and test data station-wise. Since the overall number of in situ stations is relatively small, namely 637, the statistical properties of surface $NO_2$ for different test sets are very likely to differ. In particular, the mean or standard deviation of surface $NO_2$

of different test sets will vary. Hence, in order to compare the quality of surface $NO_2$ predictions on different test sets, it is reasonable to use error measures that are more robust or even insensitive against different data distributions.

In order to ensure better comparability of performances of a model on different test sets, one should not use absolute performance measures such as the Mean Absolute Error or Root Mean Square Error, since they depend on the scale of the

different test sets.



At first glance, it seems to be reasonable to consider the Mean Percentage Error

$$\text{MPE}(x^\dagger, x) = \sum_{i=1}^{n} \frac{|x_i^\dagger - x_i|}{|x_i^\dagger|}.$$

The reason why the Mean Percentage Error enables comparing performances on different test sets is the following property: For every $c \in \mathbb{R}^n$ with $c_i \neq 0$ it holds that

$\text{MPE}(cx^\dagger, cx) = \text{MPE}(x^\dagger, x),$

where $cx^\dagger$ denotes point-wise multiplication. However, since lots of in situ measurements $x_i^\dagger$ are very close to or equal to zero, the Mean Percentage Error becomes unstable. As a trade-off, we will consider performance measures $E(x^\dagger, x)$ that are *scale-insensitive*, i.e. for every $\lambda \in \mathbb{R} \setminus \{0\}$ it holds that

$$E(\lambda x^\dagger, \lambda x) = E(x^\dagger, x).$$

**Normalized Mean Absolute Error (NMAE):**

$$\text{NMAE}(x^\dagger, x) = \frac{\sum\limits_{i=1}^{n} |x_i^\dagger - x_i|}{\sum\limits_{i=1}^{n} |x_i^\dagger|},$$

so the NMAE is just the Mean Absolute Error divided by the mean absolute value of the ground truth $x^\dagger$. If normalization by the standard deviation of $x^\dagger$ instead of its mean was considered, this would lead to a measure similar to the Coefficient of Determination $R^2$, see Appendix A. Note that in contrast to the Mean Absolute Error, NMAE is scale-insensitive. Similarly,

we define the

**Normalized Mean Square Error (NMSE):**

$$\text{NMSE}(x^\dagger, x) = \frac{\sum\limits_{i=1}^{n} |x_i^\dagger - x_i|^2}{\sum\limits_{i=1}^{n} |x_i^\dagger|^2}.$$

**Pearson correlation coefficient** ($C$)**:** Whenever we talk about the correlation between $x^\dagger$ and $x$, we mean the Pearson correlation coefficient, which is defined as

$C(x^\dagger, x) = \dfrac{\text{cov}(x^\dagger, x)}{\sigma(x^\dagger)\sigma(x)},$

where $\text{cov}(x^\dagger, x)$ denotes the covariance between $x^\dagger$ and $x$ and $\sigma(x^\dagger), \sigma(x)$ are the standard deviations of $x^\dagger$ and $x$, respectively. It is to be noted that this is not a performance measure in the sense that $x^\dagger = x$ if and only if $C(x^\dagger, x) = 1$. Nevertheless, it quantifies the linear relationsship between $x$ and $x^\dagger$. Furthermore, it is frequently used in the literature which is the reason why we consider it in our work, too.

We have considered two further scale-insensitive performance measures, the Coefficient of Determination ($R^2$) and the Index of Agreement (IOA), which are defined in Appendix A.



## 4 Machine learning models of consideration

As mentioned in the introduction, numerous machine learning models have been considered for predicting surface $NO_2$ in the literature. Examining the benefit of time-contiguous input features for all different models would be beyond the scope of 
this research. This is because fair comparisons require individual hyperparameter tuning for the models with different time-contiguity of the input features. Therefore, we restrict our attention to one approach, that has, on the one hand, performed well in the literature, and on the other hand, has not many hyperparameters to tune. If there were lots of hyperparameters to be tuned and the model's performance was very sensitive to the choice of these hyperparameters, there would be the risk that better performance was only achieved due to better hyperparameter tuning. In this study, we are going to use a Random Forest 
regressor, which we describe in Sect. 4.2 and present the selected hyperparameters. As a reference we consider a simple linear regression approach, which we recap first in the next section.

### 4.1 Linear regression

Although it has already been shown, e.g. by Ghahremanloo et al. (2021), that linear regression models are not the best for predicting surface $NO_2$, we consider an Ordinary Least Squared regressor as a reference in our study. Mainly because it has 
no tunable hyperparameters, such as regularization parameters, or architecture parameters like those in neural networks (e.g. number of layers, width of layers, activation functions, skip connections, etc.). Thus, it provides a clear view on the question whether time-contiguous inputs are beneficial for this linear regression model. During this study, we used the Ordinary Least Squares regression model provided by the Python *scikit-learn* package (version 1.2.2, Pedregosa et al. (2011)). In our case of predicting surface $NO_2$ from time-contiguous inputs, the linear regression model is a parameterized function

$$\varphi_\theta : \mathbb{R}^{pk} \longrightarrow \mathbb{R}$$
$$y \longmapsto Ay + b,$$

where $y = \big(f(z,t), ... f(z,t-k+1)\big)^T$ is some (time-contiguous) feature vector defined in Eq. (3), $A$ is a $1 \times pk$ matrix and $b \in \mathbb{R}$ some bias term. Let $(y_n, s_n)_{n=1}^N$ be some training data, where $y_n$ is some feature vector at location $z_n$ and time $t_n$, and $s_n$ the corresponding in situ measurement of surface $NO_2$ at time $t_n$. Then training $\varphi_\theta$ means to search for some parameter 
$\theta = (A, b)$ that solves the minimization problem

$$\min_\theta \sum_n |\varphi_\theta(y_n) - s_n|^2,$$

We choose to minimize the squared error since the computation time is much lower compared to other losses such as the absolute error.

### 4.2 Random Forests

There are two main reasons why Random Forests, a machine learning model originally proposed by Breiman (2001), are considered within this research. First, they have already proven to be powerful for predicting surface $NO_2$ in various studies;





see, for example, Di et al. (2020), Ghahremanloo et al. (2021), Li et al. (2022), Huang et al. (2023) on OMI and TROPOMI data, and Yang et al. (2023) on GEMS data. Second, the studies Probst et al. (2018) and Probst et al. (2019) suggest that Random Forests are less tunable compared to other machine learning approaches. "Tunable" in the sense of how much the performance of a Random Forest with typical default hyperparameters can be enhanced by adjusting (tuning) these hyperparameters. As discussed before, this reduces the risk of drawing incorrect conclusions about the benefit of using time-contiguous inputs.

In fact, according to Probst et al. (2018), mainly four hyperparameters empirically determine the performance of a Random Forest:

- The number of randomly drawn features considered at every split of a tree. In the Python scikit-learn software package (version 1.2.2, Pedregosa et al. (2011)) that we use for this study it is called `max_features`. However, in several other software packages it is denoted by `mtry`.

- The number of trees the random forest is built of. In scikit-learn it is called `n_estimators`. To be precise, it is not actually a hyperparameter, since more trees are in general more advantageous, see e.g. Genuer et al. (2008) or Scornet (2017).

- The maximal number of (randomly drawn) data samples from the training set that is used for the construction of an individual tree, denoted by `max_samples` in scikit-learn.

- The minimal number of observations that land in a leaf node during the training process. In scikit-learn it is called `min_samples_leaf`.

In their experiments Probst et al. (2018) observed that `max_features` had the biggest influence on the performance and the influence of `max_samples` and `min_samples_leaf` were smaller. This is why during hyperparameter tuning, we mainly focus on `max_features`, but also consider different values for `max_samples`. Regarding `max_samples`, we consider values between $50\%$ and $100\%$ of the size of the training dataset. On the other hand, for `max_features` values between 1 and $(pk)/3$ are considered, where $pk$ is the number of inputs for the model, so the dimension of the time-contiguous feature vector in Eq. (3). The value $(pk)/3$ is the default value of scikit-learn. Genuer et al. (2008) suggested $\sqrt{pk}$ for problems in which the number of data points is much larger than the number of input features $pk$, which clearly is the case in our study (hundred thousands of data points versus less than ninety input features). As $pk \geq 17$ the value $\sqrt{pk}$ is always within the considered interval during optimization. In fact, $\sqrt{pk}$ turns out to be quite close to the optimal choice in our hyperparameter study. Regarding `min_samples_leaf`, we inspect two typical default values, namely 1 and 5. Due to the rule "the more, the better" for the number of trees (`n_estimators`) in the forest, we use 8000 trees while tuning the other hyperparameters. hyperparameter selection is made according to the spatially cross validated (ten splits) NMSE, leading to `max_features` $= 2, 3, 3, 3, 4$ for time-contiguity $k = 1, 2, 3, 4, 5$, and further `min_samples_leaf` $= 5$ as well as for `max_samples` using $100\%$ of the size of the training data. All remaining hyperparameters are always set to the default values within scikit-learn.

With 8000 trees, we chose a very high value for the number of trees, which might need an explanation. The good message first: Comparable results can be obtained with far less trees in the forest. However, for hyperparameter tuning as well as a clearer





insight into the benefit of time-contiguous features, it is reasonable to choose a large number of trees, which we illustrate in the following: The Random Forest algorithm in scikit-learn is not deterministic, meaning that if the model gets trained on the same training data multiple times, the trained forests will differ from each other, also causing the performance on the respective test dataset to vary. However, we observe that with a higher number of trees in the forest the variance of the performance decreases for all considered performance measures. In Figure C1 in Appendix C, we illustrate this effect using a single split

into training and test stations. Two Random Forests, one with 30 trees and the other with 8000 trees, are each trained and tested 20 times on the same data, similar to Experiment 2, but with 20 repetitions of the same split instead of 60 different splits. We observe that with 30 trees the scores on the test data, such as Pearson correlation, NMSE or NMAE, exhibit some variance. In contrast, there is barely any variance in case of 8000 trees. This has the advantage that for each split into training and test stations, the Random Forest only needs to be trained once to get an interpretable result. Thereby, it also reduces the risk of

choosing non optimal hyperparameters. Therefore, during all experiments, we set the number of trees to a very large number (n_estimators $= 8000$) to stabilize the non-deterministic behavior of training a Random Forest. Note that stability is probably achieved with far less than 8000 trees. However, in order to reduce the bias from the observation above for a single split and single choice of hyperparameters, we choose a very large number that is still manageable regarding storage and computation time.

## 445   5   Results

Before presenting the results and starting the discussion, it is important to recall that for a given spatial split into training and test in situ stations, training or testing a machine learning model on the dataset $D_{N,k}$ means that only the data points corresponding to the training or test station locations are used, respectively. Furthermore, for fixed $N$, the in situ measurements $s(z,t)$ of surface $NO_2$ (ground truth) that are to be predicted in $D_{N,k}$ are exactly the same for all different $k$. Further, recall that

$D_{N,k}$ can be thought of as the set of those data points, for which also measurements at all $N-1$ previous hours are guaranteed to be available, but only $k-1$ are added to the time-contiguous feature vector in Eq. (3).

In the following discussion of the experiments, introduced in Sect. 3.2, we will focus exclusively on the results when $D_{4,k}$ is used for constructing test datasets, i.e., for $N = 4$ only. This is because we observe similar benefit from larger time-contiguity $k$ when evaluating the machine learning models' performance on $D_{N,k}$ for $N \in \{2,3,5\}$. As a further example, we provide

detailed results for $N = 2$ in Fig. C2 and Fig. C3 in Appendix C.

### 5.1   Experiment 1: Time-contiguous inputs provide additional information

In Experiment 1, we train linear regression models and Random Forests on $D_{4,k}$ for different time-contiguities $k \in \{1,...,4\}$ of the input features. The test performances of these models are evaluated via six times spatial 10-fold cross validation and are illustrated in Fig. 2 (b) and Fig. 3 (b), respectively. Specifically, we show average Pearson correlation, NMSE and NMAE over

all 60 splits into training and test stations. We observe that, on average, both linear regression and Random Forests benefit from larger time-contiguity $k$ regarding all considered performance measures. For example, the average correlation strictly increases





from 0.702 for $k = 1$ to 0.737 for $k = 4$ in the case of linear regression, and for Random Forests, it increases from 0.802 to 0.817. Further, the average NMSE decreases from 0.196 to 0.171 for linear regression and from 0.139 to 0.129 for Random Forests. Therefore, both models benefit from larger time-contiguity, but linear regression shows a greater improvement, which

is expected as it cannot model non-linear effects. Furthermore, we observe that the larger $k$, the smaller the improvement compared to the case $k - 1$, which is to be expected since input features at time $t - k$ presumably have a decreasing impact on surface NO$_2$ at time $t$ for larger $k$.

Although the visualization of average performances suggests an overall trend, it does not clearly indicate whether larger time-contiguities ($k > 1$) consistently improve performance across all 60 station splits during cross validation compared to

$k = 1$. However, we have found that this improvement holds true for all 60 station splits. The performance curves for individual splits are more or less parallel to the average curve. In Fig. 2 (a) and Fig. 3 (a) we illustrate this for exemplary station splits, where only five splits are shown for better visibility. To quantify the gain in performance for individual splits between using time-contiguity $k = 1$ and larger time-contiguities $k > 1$, we proceed as follows: For a given test dataset, let $E_k$ be the test performance (e.g. correlation) achieved by the model using time-contiguity $k$ for its inputs. We define the *performance gain* of

this model over the case with no time-contiguity $k = 1$ in Experiment 1 as

$$\frac{E_1 - E_k}{E_1 - E_{\mathrm{opt}}}, \tag{4}$$

where $E_{\mathrm{opt}}$ is the optimal value of the respective performance measure, e.g., $E_{\mathrm{opt}} = 1$ for the Pearson correlation or $E_{\mathrm{opt}} = 0$ for NMSE and NMAE. The average performance gains for the cases $k \in \{2, 3, 4\}$ compared to $k = 1$ are depicted in Fig. 2 (c) and 3 (c) for linear regression and Random Forests, respectively. In both cases and for all performance measures, the highest

average performance gain is achieved with $k = 4$. Specifically, linear regression models achieve average performance gains of 15.2% in correlation, 13.0% in NMSE and 7.7% in NMAE, whereas Random Forests achieve gains of around 7.8%, 7.0% and 4.7%, respectively. It is noteworthy that for linear regression, across all 60 splits the performance gain is at least around 12.0% in correlation, 10.0% in NMSE and 6.1% in NMAE. On the other hand, Random Forests achieve at least performance gains of 4.6%, 4.0% and 3.1%, respectively. Therefore, utilizing larger time-contiguity consistently provided beneficial additional

information for both linear regression and Random Forest models.

Additionally, for $k = 1$ and the best time-contiguity $k = 4$, we examine for each split the orthogonal regression curve between the models' predictions and ground truth measurements of surface NO$_2$ on the corresponding test dataset. For a fixed split, this is illustrated as a two-dimensional histogram in the first row of Fig. 4 for linear regression and in Fig. 5 for Random Forests. Although the histograms are restricted to surface NO$_2$ and predictions between $0\,\mu\mathrm{g\,m^{-3}}$ and $40\,\mu\mathrm{g\,m^{-3}}$ for better visibility, all

data points are taken into account for determining the orthogonal regression curve. It becomes evident that both the slope and the bias of the orthogonal regression curve improve for $k = 4$ (column (b)) compared to $k = 1$ (column (a)), where improvement means that the slope gets closer to 1 and the bias closer to 0. In the second row of these figures, we plot the mean orthogonal regression curve, which represents the mean slope and mean bias of all 60 orthogonal regression curves. An upper bound for all these curves is represented by the line with the maximal slope and bias across all splits (note that maximal slope and bias might

not occur for the same split). Similarly, a lower bound is obtained and both bounds are shown within the same plots. Both the





mean orthogonal regression curve and the upper and lower bounds improved for $k = 4$ for both linear regression and Random Forests. However, the improvement is larger for the linear regression models, which is consistent to the previous discussion on performance measures, such as the NMSE.

We want to stress another observation: Having a look at the upper and lower bounds for the orthogonal regression curves, we see that all slopes are smaller than $1$, whereas all biases are positive. Further, there is quite some gap towards the identity line. Regarding the latter, one possible explanation could be that spatially splitting the dataset into training and test sets causes a large difference in the statistical properties of the training and test sets. Simply, because overall there are just $637$ different in situ stations available, so that the Law of Large Numbers may not yet apply well when sampling $10\%$ of test stations. However, this does not explain why the slopes and biases are not more symmetrically distributed around the slope $1$ and bias $0$. Studying the impact of the number of available in situ stations and their locations on the slopes and biases of these orthogonal regression curves will be an interesting task for future work.



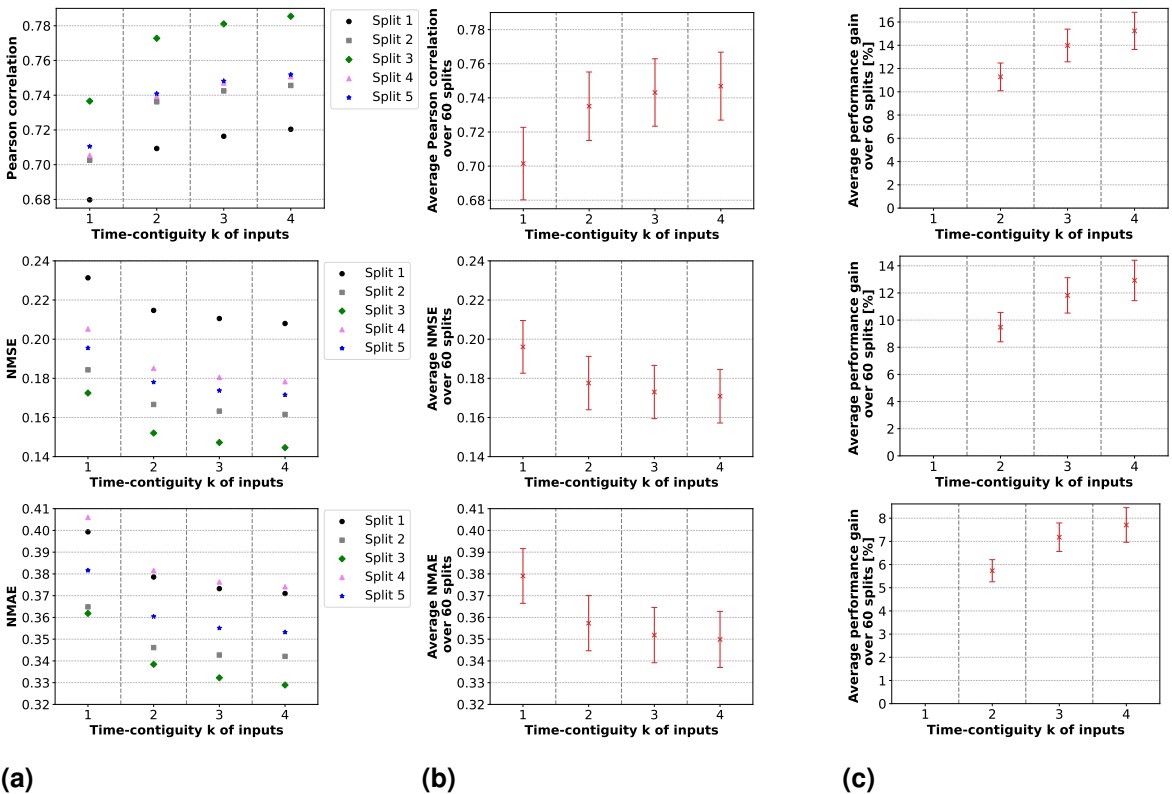

**Figure 2.** Linear regression models have been trained and tested on datasets $D_{4,k}$ for 60 different splits into training and test stations; with different time-contiguity $k$ of the input features. In column (a), performances on test sets are shown for five exemplary station splits, w.r.t. three performance measures. Column (b) shows the average performance over all 60 splits, errorbars illustrating the standard deviation. Column (c) shows the average performance gain relative to the case $k = 1$, see Eq. (4) for the definition of performance gain. Across each row the same performance measure is considered. The exact values in (b) can be found in Table B2, columns $D_{4,1}$ to $D_{4,4}$.



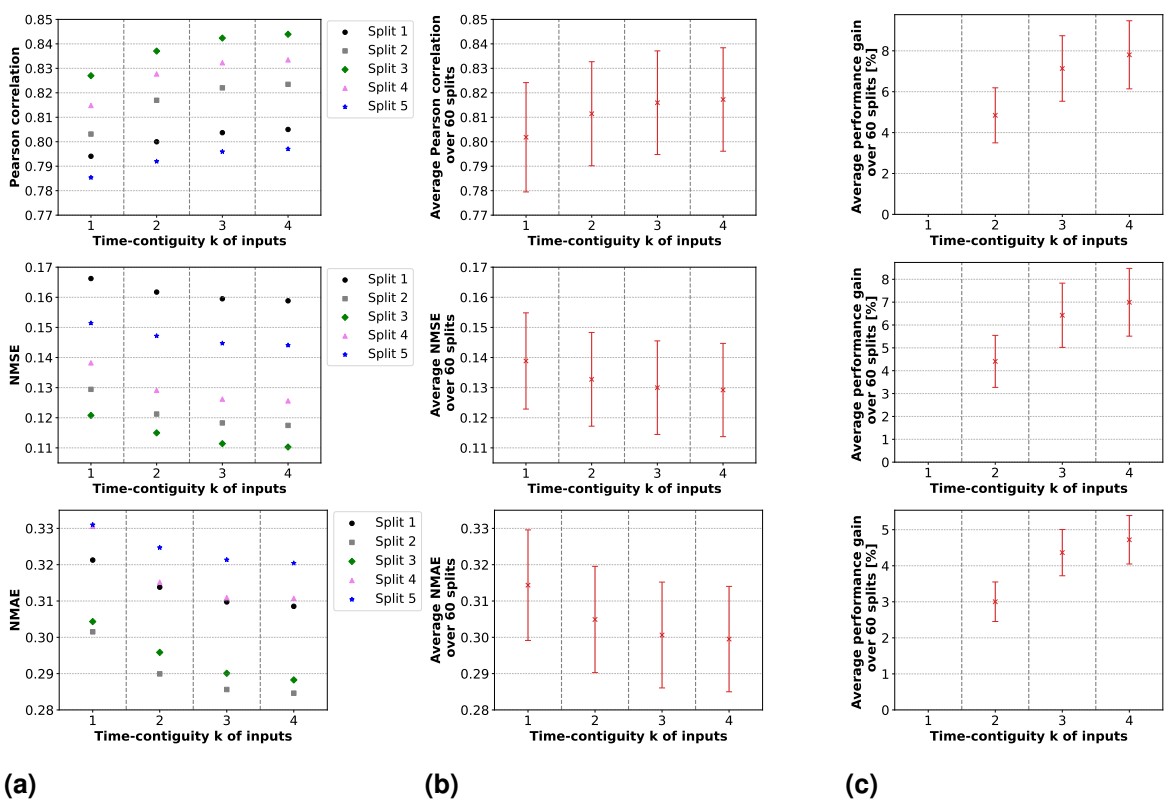

**(a)**          **(b)**          **(c)**

**Figure 3.** Same as Fig. 2, but for Random Forests: They have been trained and tested on datasets $D_{4,k}$ for 60 different splits into training and test stations; with different time-contiguity $k$ of the input features. In column (a), performances on test sets are shown for five exemplary station splits, w.r.t. three performance measures. Column (b) shows the average performance over all 60 splits, errorbars illustrating the standard deviation. Column (c) shows the average performance gain relative to the case $k = 1$, see Eq. (4) for the definition of performance gain. Across each row the same performance measure is considered. The exact values in (b) can be found in Table B3.





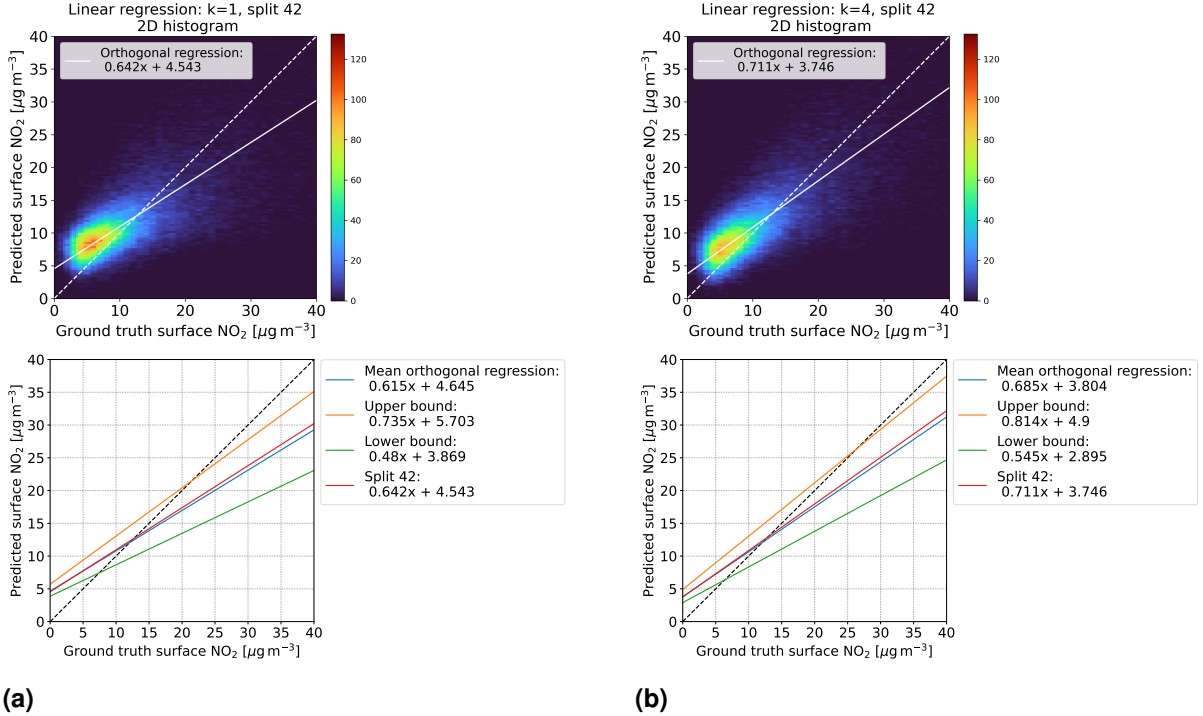

(a)                                                              (b)

**Figure 4.** Linear regression models were trained on $D_{4,k}$ with time-contiguities $k = 1$ (column (a)) and $k = 4$ (column (b)). First row: For a fixed split (number 42) into training and test stations, the models' predictions on the corresponding test set $D_{4,k}$ are compared with in situ measurements of surface $NO_2$ (ground truth) in a two-dimensional histogram. Second row: For all 60 station splits, orthogonal regression has been considered between predicted and ground truth surface $NO_2$. Mean orthogonal regression refers to the line of average slope and bias over all 60 regression lines (blue line). Also the regression line for the example in the first row is shown (red line)





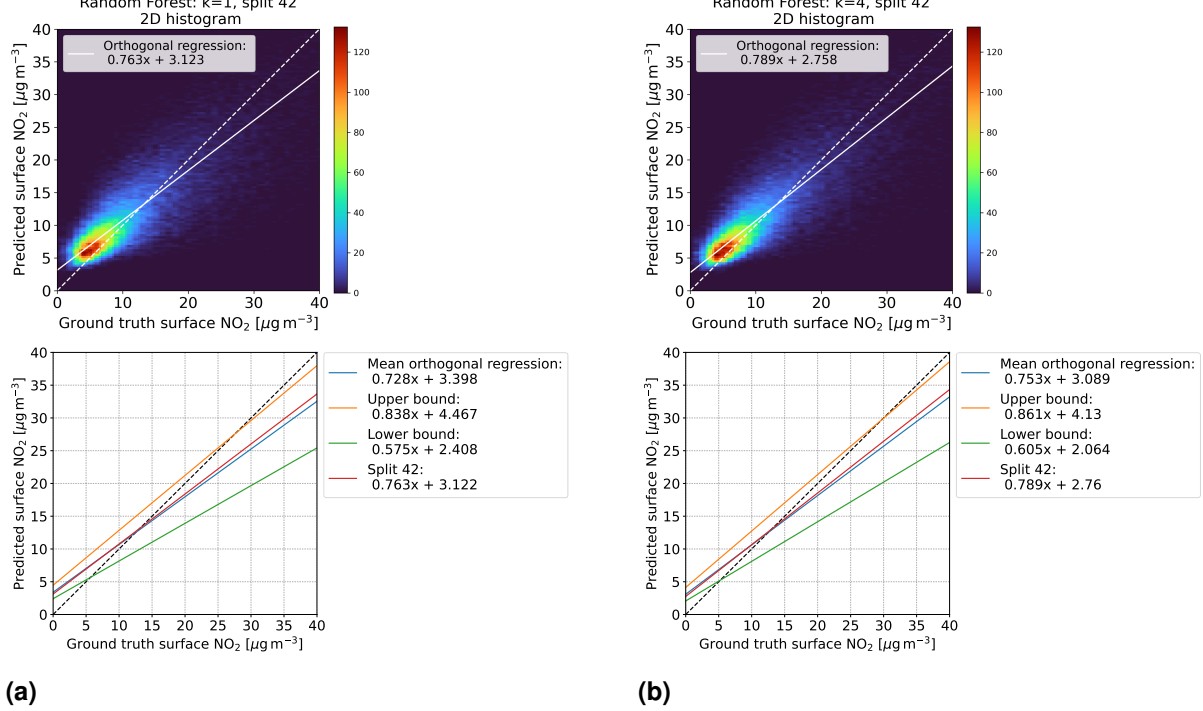

**(a)**                    **(b)**

**Figure 5.** Same as Fig. 4, but for Random Forests: They were trained on $D_{4,k}$ with time-contiguities $k = 1$ (column (a)) and $k = 4$ (column (b)). First row: For a fixed split (number 42) into training and test stations, the models' predictions on the corresponding test set $D_{4,k}$ are compared with in situ measurements of surface $NO_2$ (ground truth) in a two-dimensional histogram. Second row: For all 60 station splits, orthogonal regression has been considered between predicted and ground truth surface $NO_2$. Mean orthogonal regression refers to the line of average slope and bias over all 60 regression lines (blue line). Also the regression line for the example in the first row is shown (red line)

## 5.2 Experiment 2: Time-contiguous inputs are beneficial in spite of a smaller dataset

In Experiment 1, the models were trained and tested on $D_{N,k}$ for fixed $N$, but different time-contiguity $k \in \{1,...,N\}$ of their input features. This means that for a fixed station split, the number of training data points was the same for all different $k$, since the size of $D_{N,k}$ only depends on $N$ (see Table 1). However, for $M \in \{k,...,N-1\}$, there would be significantly more data points available in $D_{M,k}$ than in $D_{N,k}$, which could be used during training. To make a fair conclusion about whether larger time-contiguity ($k > 1$) in the models' input is more beneficial compared to time-contiguity $k = 1$, we need to consider that for $k = 1$, one can also train on these larger datasets. It is to be noted that we have also considered training on smaller datasets, so on $D_{M,k}$ with $M > N$. However, non-competitive results were obtained for Random Forests in these cases. Also for linear regression performances were worse, but with some exceptions regarding the NMAE, see Fig. C2 in Appendix C. This is why we restrict the following discussion to training on larger datasets ($M \leq N$) only.



Focusing again on the test case $N = 4$, we compare the performance on test sets in $D_{4,k}$ of models trained on larger datasets $D_{M,k}$ for all $M \in \{k, ..., 4\}$ and all $k \in \{1, ..., 4\}$. Note that for $M = 4$ this is just the setting of Experiment 1. Altogether, these are ten different linear regression and ten Random Forest models used for making predictions of the same ground truths in the split-dependent test sets $D_{N,k}$.

Average performance measures from spatial cross validation are shown in Fig. 6 (a) for linear regression and in Fig. 7 (a) for Random Forests. We observe that when training with time-contiguity $k = 1$, so on $D_{M,1}$, best results are obtained for $M = 4$. In other words, there is no improvement on the test set $D_{4,1}$ if training is done on the larger datasets ($M \in \{1, 2, 3\}$). There is one exception for Random Forests with the Pearson correlation, where training on $D_{3,1}$ yields slightly better results on average compared to training on $D_{4,1}$. However, this difference is quite small, as shown in Fig. 7 (a). Moreover, for all performance measures, best performance across all ten different training cases is achieved by the models trained on $D_{4,4}$ with time-contiguity $k = 4$. Note that this is one of the training settings already considered in Experiment 1.

For individual splits, we consider the performance gains that models with time-contiguity $k > 1$ achieve compared to models with no time-contiguity ($k = 1$). Since, in contrast to Experiment 1, we are now dealing with four different training cases for $k = 1$, we slightly adapt the definition of performance gains from Eq. (4): For a given split into training and test stations and fixed $N$, let $E_{M,k}$ be the test performance (e.g. correlation) on $D_{N,k}$ achieved by a model trained on $D_{M,k}$. We define the performance gain achieved by this model in Experiment 2 by

$$\min \left\{ \frac{E_{P,1} - E_{M,k}}{E_{P,1} - E_{\text{opt}}} : P \in \{1, ..., 5\} \right\}. \tag{5}$$

In other words, for each split, the performance gain is always computed with respect to the best model trained without time-contiguity ($k = 1$).

Average performance gains are depicted in Fig. 6 (b) and Fig. 7 (b), which only slightly differ from those in Experiment 1, as models trained on $D_{4,1}$ are better, on average, than models trained on $D_{M,1}$. Linear regression models trained with $k = 4$ still achieve performance gains of $15.0\%$ in correlation, $12.8\%$ in NMSE and $6.6\%$ in NMAE, whereas Random Forests achieve average gains of around $7.3\%$, $6.6\%$ and $4.7\%$, respectively. Again, we observe that improvements over $k = 1$ are not only true in average, but also for each individual split: Figure 6 (c) and Fig. 7 (c) show the minimal performance gains over all 60 splits. It shows that linear regression models for $k = 4$ always achieve at least an improvement of $11.7\%$ in correlation, $9.1\%$ in NMSE and $4.4\%$ in NMAE. Random Forests achieve at least gains of $2.5\%$, $3.0\%$ and $3.1\%$, respectively. Hence, models with larger time-contiguity $k > 1$ provide reliable and statistically significant improvements (w.r.t. the performance measures) over models with no time-contiguity ($k = 1$). Similar observations are made for the Coefficient of Determination and the Index of Agreement, two further performance measures. Definitions can be found in Appendix A and achieved performances in Tables B2 and B3 in Appendix B.

So far, we discussed the test case $N = 4$ in detail. In the remainder of this section, we shortly summarize our similar observations for general $N \in \{2, 3, 4, 5\}$: For all $N$, we observed that best test performances on $D_{N,k}$ are achieved when training on $D_{N,N}$, so with time-contiguity $k = N$. If $N = 5$, we observed that there is barely any difference between training on $D_{5,5}$ or on $D_{4,4}$, which implies that it is not required to use larger time-contiguity than $k = 4$. Also for general test case $N$,





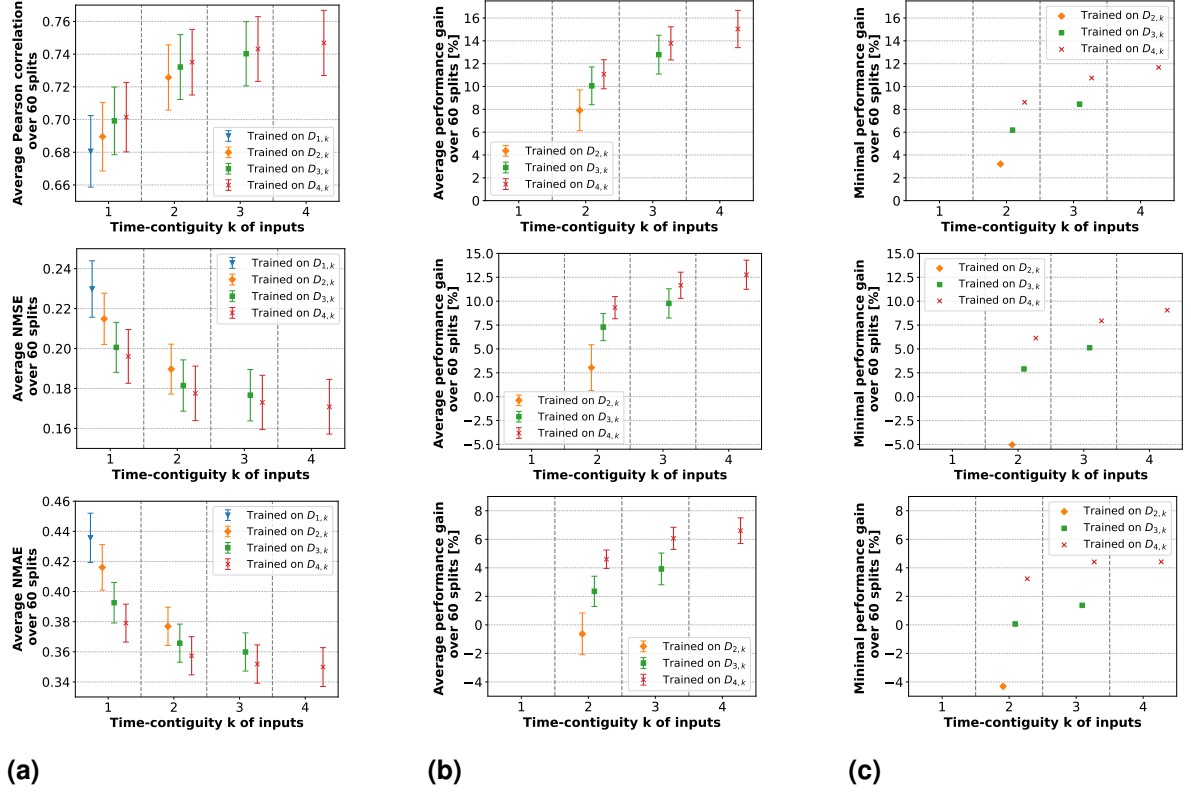

**Figure 6.** Linear regression models have been trained on $D_{M,k}$ for $M \leq 4$ with different time-contiguities $k$. Performance on $D_{4,k}$ has been evaluated by six times 10-fold spatial cross validation. Column (a) shows the average performance over all 60 station splits for three performance measures. Column (b) shows the average performance gain relative to the best case of $k = 1$, see Eq. (5) for the definition of performance gain. Errorbars illustrate the standard deviation. Column (c) shows the minimal performance gain. Across each row the same performance measure is considered. The exact values in (a) and (b) can be found in Table B2.

models trained with time-contiguity $k > 1$ achieve reliable performance gains over models with $k = 1$. Results for the test case $D_{2,k}$ are illustrated in Fig. C2 and C3 in Appendix C.

Altogether, our findings demonstrate that it is indeed reliably beneficial to use time-contiguous input features for predicting surface $NO_2$, in spite of a smaller available training dataset, which answers our main research question. As a rule of thumb:

Consider the case that surface $NO_2$ is to be predicted at some location and time, at which input features are also available at $j \geq 1$ previous hours. Then use $j' = \min\{3, j\}$ of them, in addition to the features at current time, as an input for a Random Forest that has been trained with time-contiguity $k = j' + 1$ on a dataset $D_{k,k}$. An interesting future task would be to inspect whether a similar rule can be observed for other machine learning approaches.



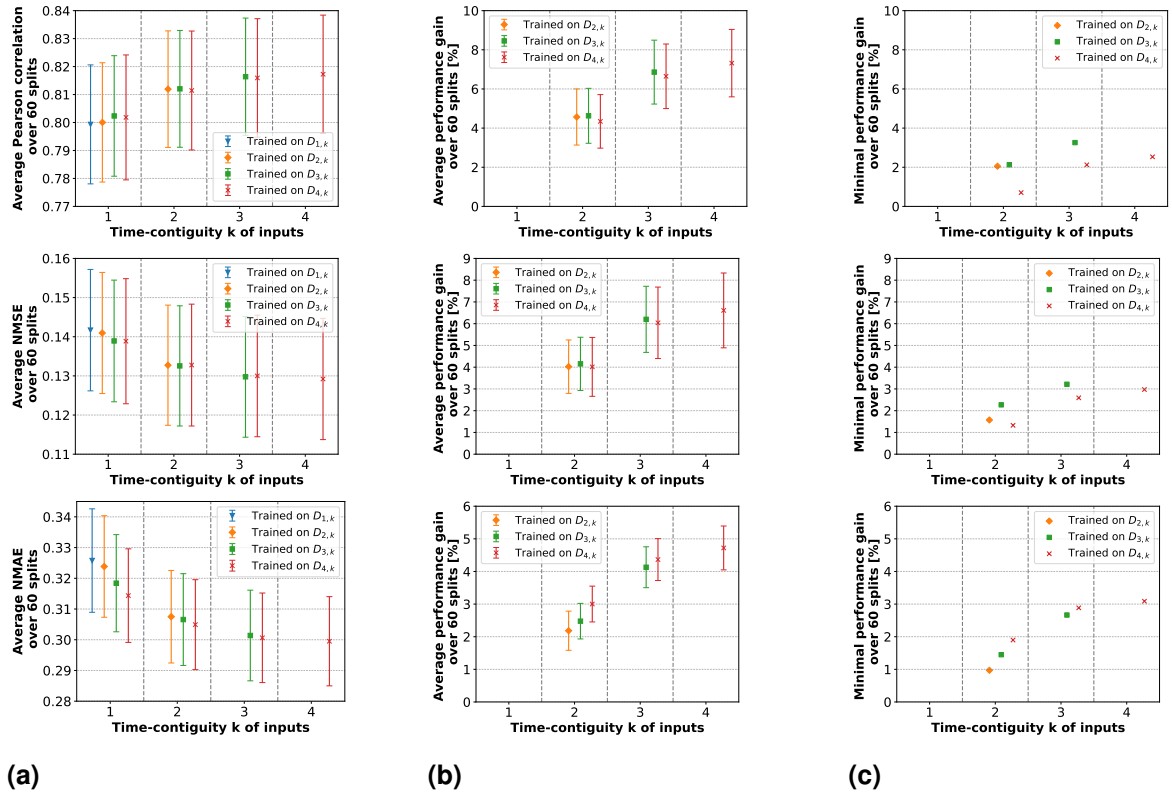

**Figure 7.** Same as Fig. 6, but for Random Forests: They have been trained on $D_{M,k}$ for $M \leq 4$ with different time-contiguities $k$. Performance on $D_{4,k}$ has been evaluated by six times 10-fold spatial cross validation. Column (a) shows the average performance over all 60 station splits for three performance measures. Column (b) shows the average performance gain relative to the best case of $k = 1$, see Eq. (5) for the definition of performance gain. Errorbars illustrate the standard deviation. Column (c) shows the minimal performance gain. Across each row the same performance measure is considered. The exact values in (a) and (b) can be found in Table B3.

### 5.3 Experiment 3: Influence of tropospheric NO$_2$ VCDs, latitude and surface height

In Experiment 3, we compare the outcomes of Experiment 2 in four different settings regarding the input of the models, as described in Sect. 3.2:

**Setting 1:** All features selected in Sect. 3.1 are included as input features, which was the setting in Experiments 1 and 2.

**Setting 2:** VCDs are excluded as an input feature.

**Setting 3:** Latitude and surface height are excluded.

**Setting 4:** VCDs, latitude and surface height are excluded.





In this section, we focus exclusively on Random Forests and discuss the test results on $D_{4,k}$ for the four different settings above.

Setting 1 has already been discussed in the previous section, where the results are illustrated in Fig. 7. Equally detailed illustrations for the remaining three settings are provided in Appendix D. A direct comparison between the four settings is
made in Fig. 8: Column (a) shows the average Pearson correlation, NMSE and NMAE achieved by Random Forests within these four settings, while column (b) displays the corresponding average performance gains. For clarity, we only include the results for the models trained on $D_{4,k}$ for different time-contiguities $k \in \{1, ..., 4\}$, excluding the models trained on larger datasets $D_{M,k}$ (similar to Experiment 1).

In Setting 3, where latitude and surface height are excluded, the models achieve similar results to those in the original Setting
1. Results are even slightly better without using these coordinates if $k > 1$. Moreover, the benefit of using time-contiguous input features is larger in Setting 3: Average performance gains, calculated with Eq. (5), achieved when training on $D_{4,k}$ are $9.3\%$ in Pearson correlation, $8.3\%$ in NMSE and $5.7\%$ in NMAE. The minimum gains across all 60 station splits are $5.4\%$, $3.7\%$ and $3.8\%$ in correlation, NMSE and NMAE, respectively (see Appendix Fig. D1). This implies that, similar to Setting 1, including time-contiguous features also provides a reliable improvement in Setting 3. This observation that coordinates are not required
as inputs to make good predictions is promising, since it presumably increases the models' chances to perform also well outside of Korea. Nevertheless, this hypothesis remains to be investigated within further research.

When excluding the tropospheric $NO_2$ VCDs (Setting 2), all performance measures decline, which is expected because the VCDs correlate the most among all input features with the surface $NO_2$ measurements. Despite this, the performances remain acceptable. For instance, with time-contiguity $k = 1$, average Pearson correlation in Setting 2 is $0.78$, whereas it is about $0.8$ in
Setting 1 and 3, when VCDs are included. Interestingly, without VCDs in Setting 2, the average performance gains achieved with larger $k$ are significantly lower: In Setting 2, the average performance gain is around $2\%$, whereas in Settings 1 and 3, it is $3.5$ and $4.5$ times larger, respectively. Consequently, for time-contiguity $k = 4$, the difference in performance is larger: Models in Setting 2 achieve an average correlation of $0.786$, while those in Settings 1 and 3 reach almost $0.82$. When tropospheric $NO_2$ VCDs, latitude and surface height are excluded in Setting 4, performances not only further weaken, but the performance gains
drop below $1\%$. In Setting 4, the average correlation is below $0.765$ for all $k$. Similar trends are observed for the NMSE and NMAE. This indicates that spatial coordinates play a more critical role when VCDs are excluded, which presumably leads to models that are less capable of generalizing to locations outside of Korea. Inspecting the connection between including VCDs and the model's ability to generalize to locations outside of Korea remains an interesting task for the future.

Furthermore, when tropospheric $NO_2$ VCDs are excluded, in both Settings 2 and 4, the use of time-contiguous inputs does
no longer provide a reliable improvement. Across the 60 station splits, the performance gain is not always positive, which can be seen in Fig. 8 (b). Due to this observation that improvements by time-contiguous inputs are only reliable when including the VCDs, the following question arises: How does it affect the performance if VCDs are treated as the only time-contiguous input feature? The experiments covering this case are illustrated in Fig. D4 in Appendix D. We observe that the average performances and average performance gains are higher if also the other features are considered to being time-contiguous. Therefore, one
future task would be to find the optimal choice of time-contiguity $k$ for each input feature individually.





At the end of this section, we show in Fig. 9 an example of how predictions of surface $NO_2$ appear on a map for the four investigated settings. We consider latitudes and longitudes within $[32°\,N, 39°\,N]$ and $[124°\,E, 132°\,E]$, respectively. GEMS tropospheric $NO_2$ VCDs on 7 April 2021 from 01:45 to 02:15 UTC are shown in column (a). We chose this time and day due to little cloud cover in the area and thus only few missing satellite observations. Predictions of surface $NO_2$ from 01:00 to

02:00 UTC made by Random Forests are shown in column (b) for Settings 1 and 3, whereas column (c) covers the settings with tropospheric $NO_2$ VCDs excluded. All models have been trained with time-contiguity $k = 4$ on $D_{4,4}$.

We observe that there is a high similarity between predictions made in Settings 1 and 3, when tropospheric $NO_2$ VCDs are included as input features. This is in agreement with our findings from Fig. 8 that in both settings similar results are achieved regarding all considered performance measures. This observation is promising, as excluding latitude and surface height reduces

the spatial bias for the model, which is to be tested in future studies. Therefore, presumably, the model's chance of making suitable predictions at different parts of the world increases. In Settings 1 and 3, the impact of the tropospheric $NO_2$ VCDs on the prediction of surface $NO_2$ is directly visible, since the hot spots of the VCDs and predictions of surface $NO_2$ are depicted at the same locations. On the other hand, when VCDs are excluded in Settings 2 and 4, these hot spots are less recognizable due to smaller contrast to their neighborhood, see column (c) of Fig. 9. In Settings 2 and 4, the predicted surface $NO_2$ has a coarser

resolution, which is to be expected as the resolution of meteorological inputs is eight times coarser compared to the VCDs. In all four settings, the contrast between the hot spots and the background of predicted surface $NO_2$ is less pronounced compared to the contrast observed in the tropospheric $NO_2$ VCDs shown in column (a). This effect is even more evident in another example from 27 February 2022, shown in Fig. 10. Notably, the predicted concentrations of surface $NO_2$ over water are only slightly smaller compared to those over land within all settings, even in regions far from the coast, such as the southeastern parts

of the maps. However, emissions over water are not expected, aside from maritime traffic. Furthermore, at some distance from the coast, no contribution from land-based emissions is expected due to the short atmospheric lifetime of $NO_2$. Consequently, both the tropospheric $NO_2$ VCDs and surface $NO_2$ concentrations should be low in these areas. Given the predicted surface concentrations of approximately $7\,\mu g\,m^{-3}$, it appears that the models have likely overestimated surface $NO_2$ concentrations in these areas over water. This aligns with the observation from Fig. 5, which shows that the models tend to overestimate low

surface $NO_2$ values. A possible explanation for this could be that the models were trained only on data from stations located on land or islands.





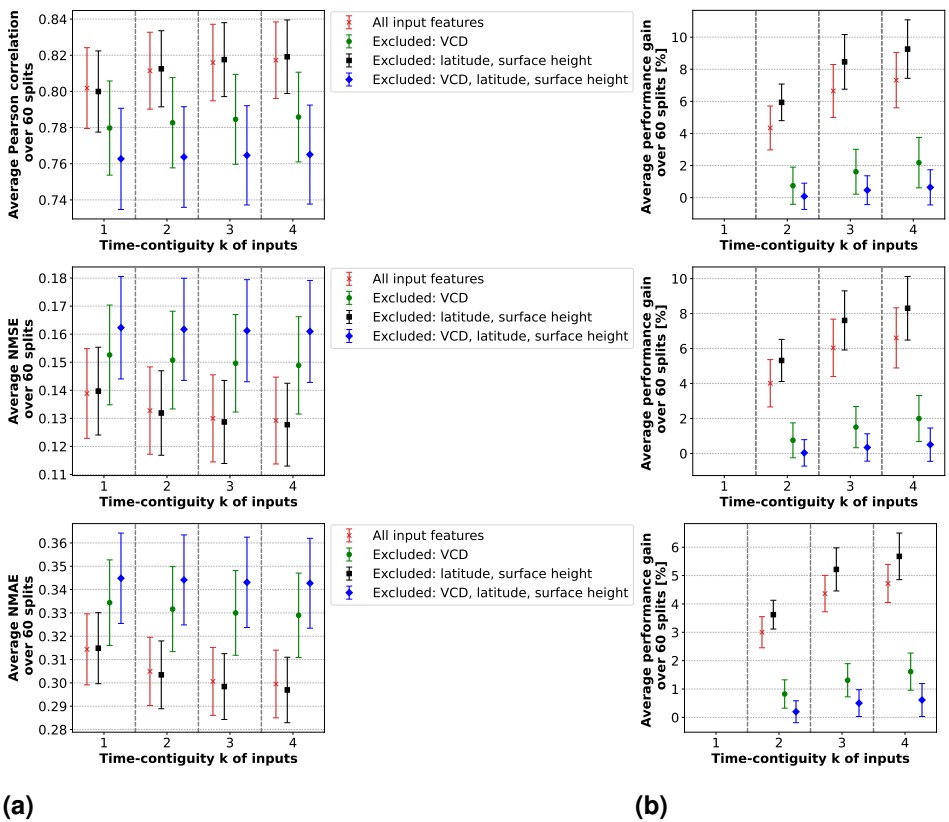

**(a)**                                                    **(b)**

**Figure 8.** In the four settings of Experiment 3 (named in the legends of the plots), Random Forests have been trained and tested on $D_{4,k}$ for different time-contiguities $k$. Performance has been evaluated by six times 10-fold spatial cross validation. Column (a) shows the average performance over all 60 station splits achieved within these four settings. Three performance measures are considered, one for each row. Errorbars illustrate the standard deviation. Column (b) shows the average performance gain relative to the best case of $k = 1$, see Eq. (5) for the definition of performance gain.



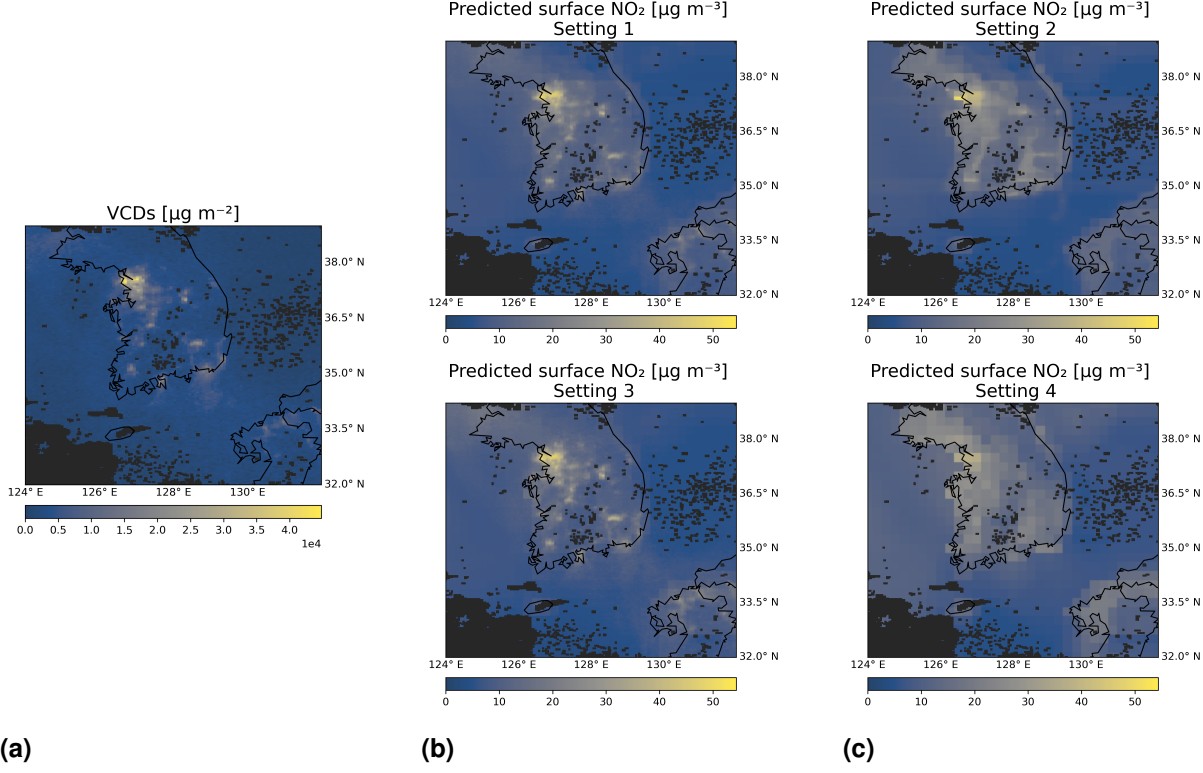

**Figure 9.** Predictions of surface NO$_2$ by Random Forests on 7 April 2021 from 01:00 to 02:00 UTC, for Settings 1-4 of Experiment 3. Column (a) shows tropospheric NO$_2$ VCDs from 01:45 to 02:15 UTC. Column (b) shows predicted surface NO$_2$ in Settings 1 and 3, when VCDs are included as an input. Column (c) shows predictions in Settings 2 and 4, when VCDs are excluded. In the second row of (b) and (c), latitude and surface height were excluded. The black mask indicates missing data, e.g. due to clouds. All models have been trained with time-contiguity $k = 4$ on $D_{4,4}$ for the same choice of training stations.





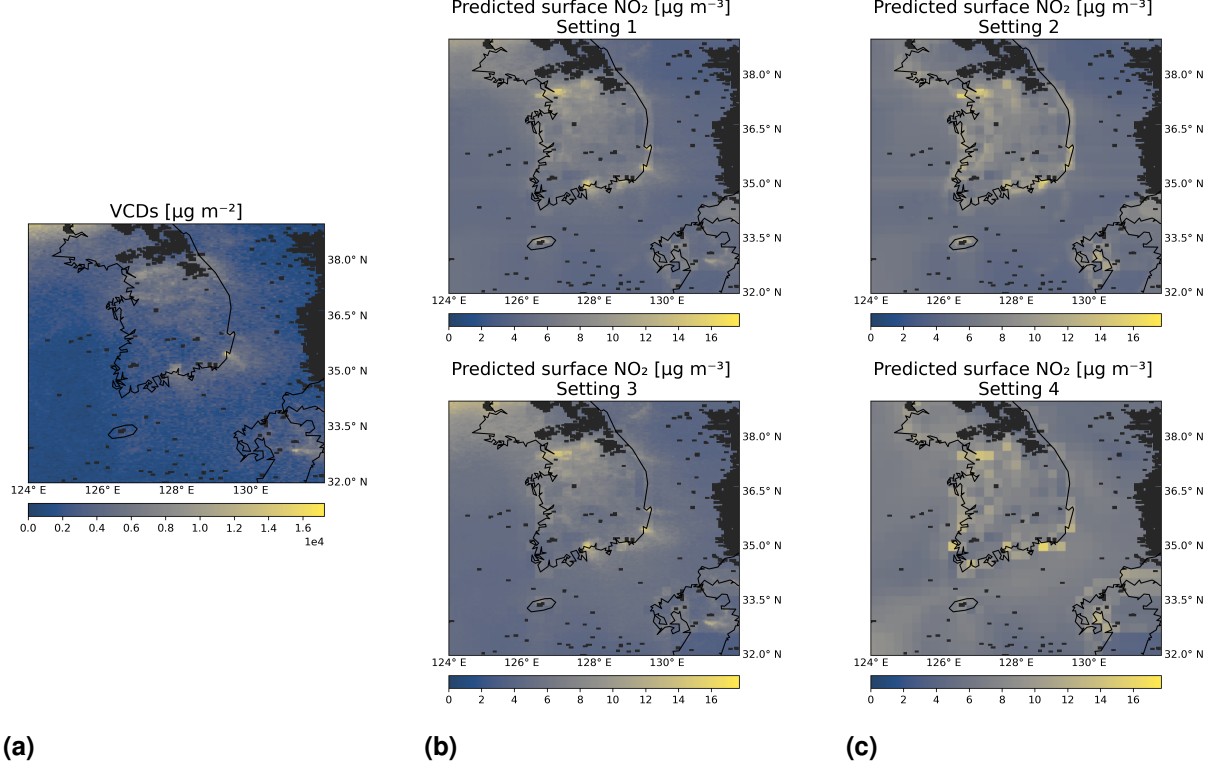

**(a)**          **(b)**          **(c)**

**Figure 10.** Same as Fig. 9, but on 27 February 2022. Column (a) shows the VCDs from 06:45 to 07:15 UTC. Columns (b) and (c) show predicted surface $NO_2$ from 06:00 to 07:00 UTC, for the four settings of Experiment 3.

## 6 Conclusions and outlook

For the first time, hourly tropospheric $NO_2$ VCDs are available due to the satellite's geostationarity of the GEMS instrument platform. To predict surface $NO_2$ levels at some time and location, we proposed to include VCDs and meteorological features also at previous hours as inputs for the machine learning models.

Our main research question was whether the considered machine learning models Random Forests and linear regression benefit from hourly time-contiguous input features for the prediction of surface $NO_2$. We observed that using time-contiguous input features led to reliable enhancements with respect to all considered performance measures, as long as tropospheric $NO_2$ VCDs were included. For Random Forests, average performance gains were between $4.5\%$ and $7.5\%$ depending on the performance measure. For linear regression models, average performance gains were larger, namely between $7\%$ and $15\%$. This is to be expected since the non-linear structure of Random Forests allows for extracting more information from non time-contiguous inputs, leading also generally to better predictions compared to linear regression models. These improvements



were reliable in the sense that positive performance gains were not only achieved on average, but for all 60 splits into training and test in situ stations during spatial cross validation. Moreover, we were able to demonstrate that performance gains were
observed despite having much fewer data points available for training models with larger time-contiguity of their inputs. As a rule of thumb, for the case that tropospheric NO$_2$ VCDs are used as an input feature, we suggest: Whenever surface NO$_2$ is to be predicted at some location and time at which input features are available at $j$ previous hours, feed them, together with features at current time, into a Random Forest that has been trained with time-contiguity $k = \min\{j+1, 4\}$ on some training dataset $D_{k,k}$, specified in Sect. 2.3. Whether this rule still applies for other machine learning models, such as Neural Networks
or Extreme Gradient Boosting, would be an interesting aspect for future studies.

Furthermore, when tropospheric NO$_2$ VCDs were included as an input of the models, we observed that latitude and surface height were not required for achieving similar performances and benefits from time-contiguous inputs. Presumably, this increases the chance that the models will provide good predictions also beyond Korea, which will be an interesting investigation for future work. If validated, this would enhance the model's flexibility and broader applicability without the requirement of
more training data, and hence larger training time, from different regions. Another task would be to decide for every input feature individually about the optimal time-contiguity, which would reduce redundancy among input features and hence could lead to better performances.

When tropospheric NO$_2$ VCDs were excluded as input features, performance worsened, but remained within an acceptable range. Additionally, we observed that the benefit of time-contiguous features was significantly reduced, and the performance
gain was no longer reliable. Specifically, across all 60 splits during spatial cross validation, benefit was not consistently observed. When both VCDs and spatial coordinates were excluded, performance decreased further. This indicates that spatial coordinates play a more critical role when VCDs are not included, which presumably leads to models that are less capable of generalizing to locations outside of Korea. Again, this motivates further research on the connection between including VCDs and the models' ability to generalize to locations outside of Korea.

*Code and data availability.* All datasets and codes are available upon request.

**Appendix A: Further performance measures**

In the following we describe further scale-insensitive performance mesaures for the gap between surface NO$_2$ measurements $x^\dagger \in \mathbb{R}^n$ and predictions $x$ made by a machine learning model.

**Coefficient of Determination (R$^2$):**

$$R^2(x^\dagger, x) = 1 - \frac{\sum_{i=1}^{n} \left| x_i^\dagger - x_i \right|}{\sum_{i=1}^{n} \left| x_i^\dagger - \overline{x}^\dagger \right|}, \quad \text{where} \quad \overline{x}^\dagger = \frac{1}{n} \sum_{i=1}^{n} x_i^\dagger.$$



Note that $R^2$ is similar to the NMAE, but normalization is by the mean absolute deviation of $x^\dagger$ instead of its mean. Further, within the literature the expression $R^2$ sometimes stands for the square of the correlation coefficient. However, in general, these definitions are not equivalent.

**Index of Agreement (IOA):**

$$\text{IOA}(x^\dagger, x) = 1 - \frac{\sum\limits_{i=1}^{n} |x_i^\dagger - x_i|^2}{\sum\limits_{i=1}^{n} \left( |\overline{x}^\dagger - x_i| + |\overline{x}^\dagger - x_i^\dagger| \right)^2},$$

where $\overline{x}^\dagger$ denotes the mean of all $x_i^\dagger$.

**Appendix B: Tables**





**Table B1.** Features considered during feature selection in Sect. 3.1. For 200 splits into training and test stations, Pearson correlation with surface NO$_2$ was computed on the training set for each available feature. Average correlations are shown in the last column.

| | Feature name | Source | Average correlation with surface NO$_2$ |
|---|---|---|---|
| Selected features | Tropospheric vertical column density of NO$_2$ | IUP-UB retrieval on GEMS data | 0.626 |
| | Latitude at center of GEMS pixel | GEMS data product | 0.149 |
| | Surface height at center of GEMS pixel | GEMS data product | −0.185 |
| | 10 metre u-component of wind | ERA5 | −0.105 |
| | 100 metre u-component of wind | ERA5 | −0.112 |
| | Instantaneous 10 metre wind gust | ERA5 | −0.237 |
| | 2 metre temperature | ERA5 | −0.252 |
| | Surface pressure | ERA5 | 0.293 |
| | Skin temperature | ERA5 | −0.226 |
| | UV visible albedo for diffuse radiation | ERA5 | 0.297 |
| | Downward UV radiation at the surface | ERA5 | −0.217 |
| | UV visible albedo for direct radiation | ERA5 | 0.283 |
| | Boundary layer height | ERA5 | −0.318 |
| | Total column water | ERA5 | −0.212 |
| | Evaporation | ERA5 | 0.239 |
| | Soil type | ERA5 | 0.163 |
| | High vegetation cover | ERA5 | −0.130 |
| Excluded features | Measuring time (hour) | Defined in Sect. 2.2 | 0.001 |
| | Longitude at center of GEMS pixel | GEMS data product | −0.054 |
| | 10 metre v-component of wind | ERA5 | 0.076 |
| | 100 metre v-component of wind | ERA5 | 0.076 |
| | Vertical integral of temperature | ERA5 | −0.009 |
| | Total column ozone | ERA5 | 0.062 |





**Table B2.** Linear regression models have been trained on $D_{N,k}$ for $N \leq 4$ with different time-contiguities $k$ and input features selected in Sect. 3.1. Performance on $D_{4,k}$ has been evaluated by six times 10-fold spatial cross validation. Five different performance measures are considered, defined in Sect. 3.3 and Appendix A. Best results are marked bold.

| | | | | | | Training datasets $D_{N,k}$ | | | | | |
| | | $D_{1,1}$ | $D_{2,1}$ | $D_{3,1}$ | $D_{4,1}$ | $D_{2,2}$ | $D_{3,2}$ | $D_{4,2}$ | $D_{3,3}$ | $D_{4,3}$ | $D_{4,4}$ |
|---|---|---|---|---|---|---|---|---|---|---|---|
| Correlation | mean | 0.6806 | 0.6895 | 0.6992 | 0.7015 | 0.7257 | 0.7321 | 0.7351 | 0.7402 | 0.7431 | **0.7469** |
| | std | 0.0219 | 0.021 | 0.0207 | 0.0212 | 0.0199 | 0.0198 | 0.0201 | 0.0196 | 0.0198 | 0.0199 |
| | mean gain [%] | - | - | - | - | 7.9109 | 10.0592 | 11.0761 | 12.7933 | 13.7819 | **15.0394** |
| | std gain [%] | - | - | - | - | 1.788 | 1.6522 | 1.2735 | 1.699 | 1.4521 | 1.6349 |
| NMSE | mean | 0.2298 | 0.2149 | 0.2006 | 0.1961 | 0.1897 | 0.1815 | 0.1776 | 0.1766 | 0.173 | **0.1709** |
| | std | 0.0141 | 0.0128 | 0.0125 | 0.0135 | 0.0125 | 0.0128 | 0.0136 | 0.0129 | 0.0136 | 0.0137 |
| | mean gain [%] | - | - | - | - | 3.0353 | 7.2854 | 9.3237 | 9.7677 | 11.6669 | **12.7688** |
| | std gain [%] | - | - | - | - | 2.3991 | 1.4194 | 1.162 | 1.5324 | 1.3681 | 1.5287 |
| NMAE | mean | 0.4357 | 0.4161 | 0.3926 | 0.3791 | 0.3769 | 0.3657 | 0.3573 | 0.3599 | 0.3519 | **0.3499** |
| | std | 0.0164 | 0.0151 | 0.0135 | 0.0126 | 0.0127 | 0.0126 | 0.0127 | 0.0127 | 0.0127 | 0.0129 |
| | mean gain [%] | - | - | - | - | −0.6329 | 2.354 | 4.6017 | 3.922 | 6.0653 | **6.6** |
| | std gain [%] | - | - | - | - | 1.464 | 1.0568 | 0.6454 | 1.1123 | 0.7738 | 0.8988 |
| $R^2$ | mean | 0.3984 | 0.4378 | 0.4754 | 0.4874 | 0.5038 | 0.5255 | 0.5359 | 0.5382 | 0.5479 | **0.5535** |
| | std | 0.0432 | 0.0361 | 0.0311 | 0.0308 | 0.0324 | 0.0305 | 0.0305 | 0.0304 | 0.0303 | 0.0306 |
| | mean gain [%] | - | - | - | - | 3.0353 | 7.2854 | 9.3237 | 9.7677 | 11.6669 | **12.7688** |
| | std gain [%] | - | - | - | - | 2.3991 | 1.4195 | 1.162 | 1.5324 | 1.3681 | 1.5287 |
| IOA | mean | 0.809 | 0.811 | 0.8096 | 0.8003 | 0.8381 | 0.8365 | 0.8283 | **0.8423** | 0.8349 | 0.8379 |
| | std | 0.0145 | 0.0149 | 0.0164 | 0.0185 | 0.0145 | 0.0156 | 0.0173 | 0.0154 | 0.017 | 0.0169 |
| | mean gain [%] | - | - | - | - | 14.0378 | 13.2159 | 8.9272 | **16.3166** | 12.3957 | 14.018 |
| | std gain [%] | - | - | - | - | 1.5684 | 2.1544 | 2.9093 | 2.2224 | 2.9518 | 2.9977 |



**Table B3.** Random Forests have been trained on $D_{N,k}$ for $N \leq 4$ with different time-contiguities $k$ and input features selected in Sect. 3.1. Performance on $D_{4,k}$ has been evaluated by six times 10-fold spatial cross validation. Five different performance measures are considered, defined in Sect. 3.3 and Appendix A. Best results are marked bold.

| | | Training datasets $D_{N,k}$ | | | | | | | | | |
| --- | --- | --- | --- | --- | --- | --- | --- | --- | --- | --- | --- |
| | | $D_{1,1}$ | $D_{2,1}$ | $D_{3,1}$ | $D_{4,1}$ | $D_{2,2}$ | $D_{3,2}$ | $D_{4,2}$ | $D_{3,3}$ | $D_{4,3}$ | $D_{4,4}$ |
| Correlation | mean | 0.7993 | 0.8 | 0.8023 | 0.8018 | 0.8119 | 0.812 | 0.8114 | 0.8164 | 0.8159 | **0.8173** |
| | std | 0.0213 | 0.0213 | 0.0216 | 0.0223 | 0.0208 | 0.0209 | 0.0213 | 0.021 | 0.0212 | 0.0211 |
| | mean gain [%] | - | - | - | - | 4.5676 | 4.6283 | 4.3439 | 6.8605 | 6.6466 | **7.3194** |
| | std gain [%] | - | - | - | - | 1.4329 | 1.4029 | 1.3676 | 1.6319 | 1.649 | 1.7219 |
| NMSE | mean | 0.1417 | 0.141 | 0.1389 | 0.1389 | 0.1327 | 0.1326 | 0.1328 | 0.1298 | 0.13 | **0.1292** |
| | std | 0.0155 | 0.0155 | 0.0155 | 0.016 | 0.0153 | 0.0154 | 0.0156 | 0.0154 | 0.0155 | 0.0155 |
| | mean gain [%] | - | - | - | - | 4.0239 | 4.153 | 4.015 | 6.2 | 6.0405 | **6.6102** |
| | std gain [%] | - | - | - | - | 1.2284 | 1.2229 | 1.3537 | 1.5193 | 1.6428 | 1.7201 |
| NMAE | mean | 0.3258 | 0.3238 | 0.3184 | 0.3144 | 0.3075 | 0.3066 | 0.3049 | 0.3014 | 0.3006 | **0.2995** |
| | std | 0.0168 | 0.0165 | 0.0158 | 0.0152 | 0.0151 | 0.0149 | 0.0146 | 0.0148 | 0.0146 | 0.0145 |
| | mean gain [%] | - | - | - | - | 2.1838 | 2.4769 | 3.0019 | 4.1298 | 4.3647 | **4.7212** |
| | std gain [%] | - | - | - | - | 0.6003 | 0.545 | 0.5486 | 0.6267 | 0.6423 | 0.6722 |
| $R^2$ | mean | 0.6301 | 0.632 | 0.6373 | 0.6375 | 0.6535 | 0.654 | 0.6534 | 0.6613 | 0.6607 | **0.6627** |
| | std | 0.0337 | 0.0337 | 0.0342 | 0.0355 | 0.0336 | 0.0338 | 0.0344 | 0.0341 | 0.0345 | 0.0344 |
| | mean gain [%] | - | - | - | - | 4.0239 | 4.153 | 4.015 | 6.2 | 6.0405 | **6.6102** |
| | std gain [%] | - | - | - | - | 1.2284 | 1.2229 | 1.3537 | 1.5193 | 1.6428 | 1.7201 |
| IOA | mean | 0.8752 | 0.8756 | 0.8768 | 0.875 | 0.8846 | 0.8846 | 0.8833 | **0.887** | 0.886 | 0.8866 |
| | std | 0.0153 | 0.0153 | 0.0155 | 0.0162 | 0.015 | 0.0151 | 0.0154 | 0.0151 | 0.0153 | 0.0153 |
| | mean gain [%] | - | - | - | - | 6.3027 | 6.3035 | 5.2754 | **8.2736** | 7.5138 | 7.9427 |
| | std gain [%] | - | - | - | - | 1.4278 | 1.498 | 1.6812 | 1.8665 | 2.0031 | 2.0893 |

**Appendix C: Additional figures for Experiment 2**



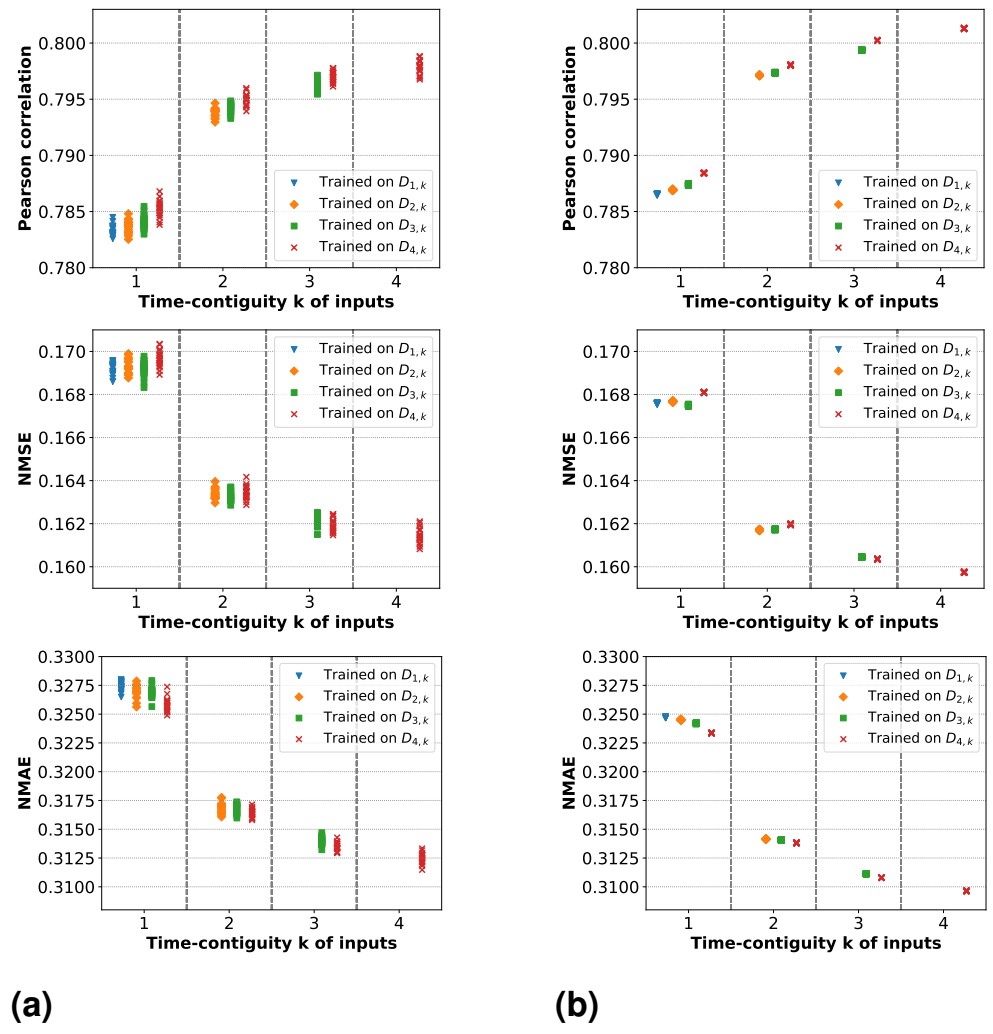

**Figure C1.** Random Forests with 30 and 8000 trees (`n_estimators`) are considered in columns (a) and (b), respectively. Training and testing have been performed 20 times for the same split into training and test stations. Testing was on the corresponding dataset $D_{4,k}$ and training on different $D_{M,k}$ for $M \leq 4$. Results for individual 20 repetitions are shown w.r.t. three performance measures.



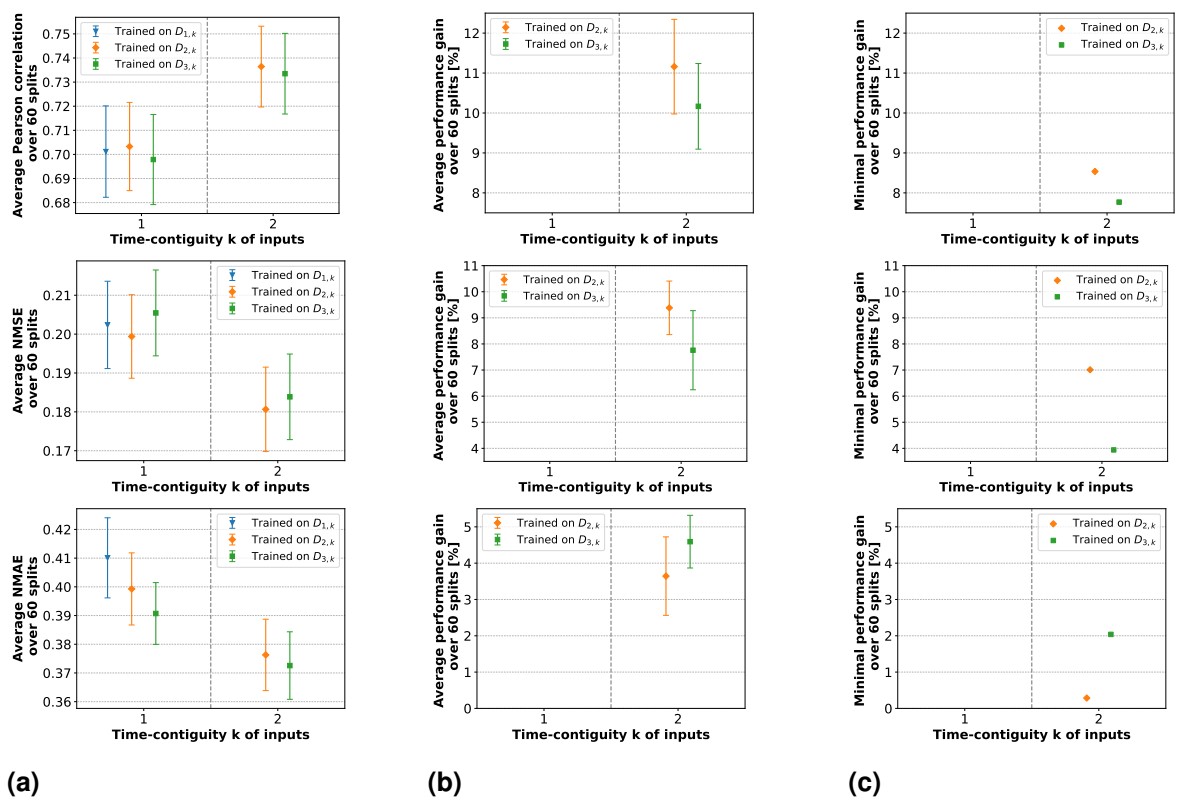

**Figure C2.** Linear regression models have been trained on $D_{M,k}$ for $M \leq 3$ with different time-contiguities $k$ and input features selected in Sect. 3.1. Performance on $D_{2,k}$ has been evaluated by six times 10-fold spatial cross validation. Column (a) shows the average performance over all 60 station splits for three performance measures. Column (b) shows the average performance gain (Eq. (5)) relative to the best case of $k = 1$. Errorbars illustrate the standard deviation. Column (c) shows the minimal performance gain. Across each row the same performance measure is considered.



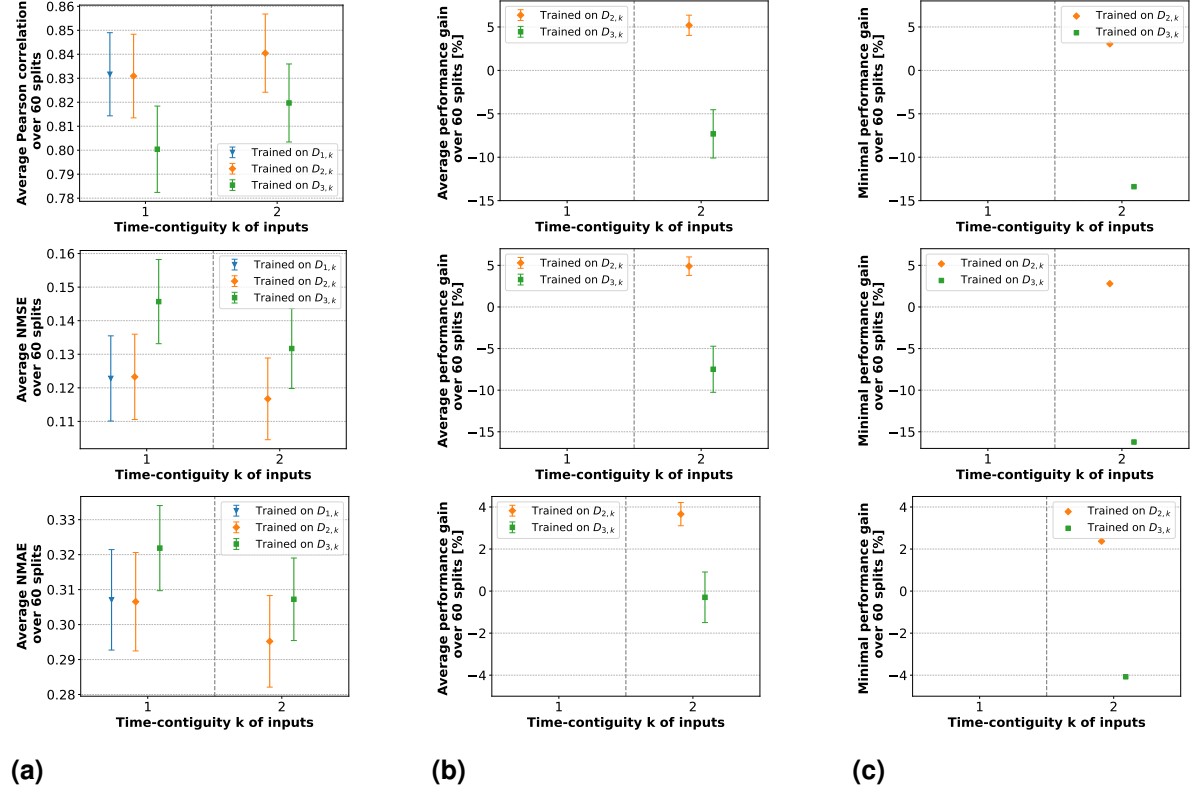

**Figure C3.** Same as Fig. C2, but for Random Forests: They have been trained on $D_{M,k}$ for $M \leq 3$ with different time-contiguities $k$ and input features selected in Sect. 3.1. Performance on $D_{2,k}$ has been evaluated by six times 10-fold spatial cross validation. Column (a) shows the average performance over all 60 station splits for three performance measures. Column (b) shows the average performance gain (Eq. (5)) relative to the best case of $k = 1$. Errorbars illustrate the standard deviation. Column (c) shows the minimal performance gain. Across each row the same performance measure is considered.

## Appendix D: Additional figures for Experiment 3



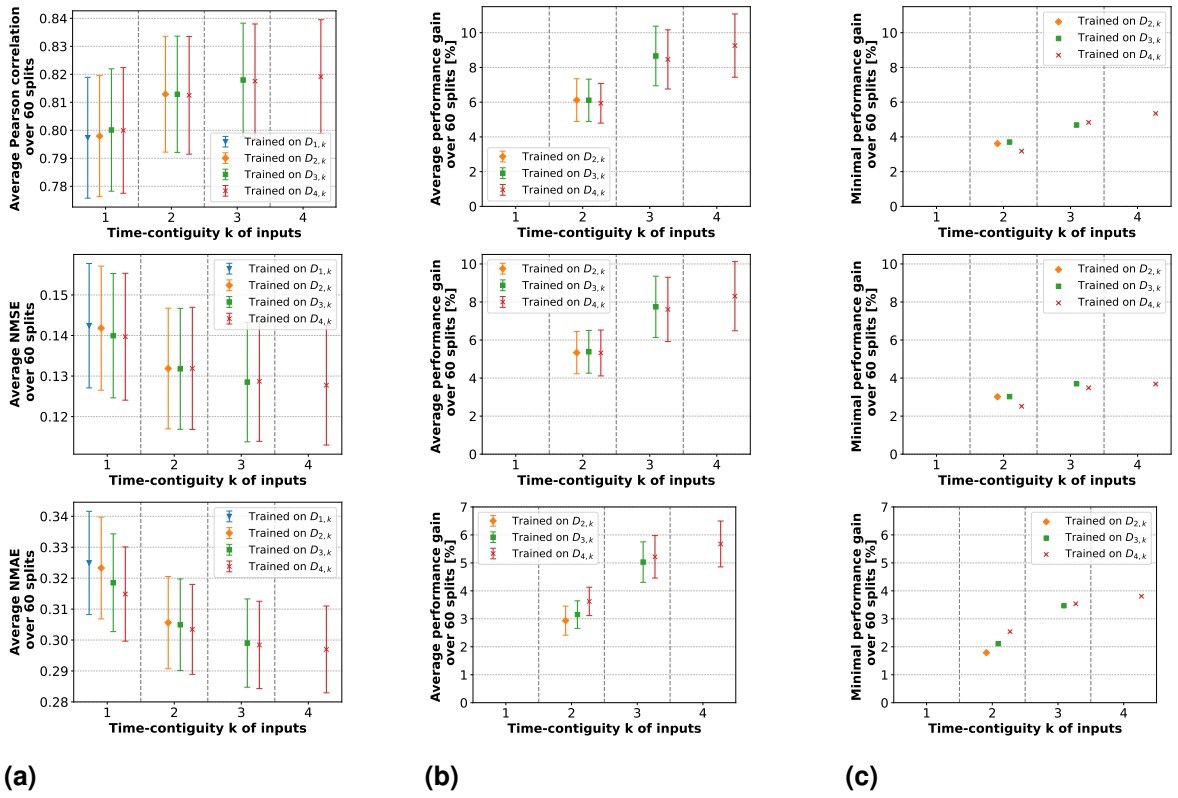

**Figure D1.** Excluded latitude and surface height from input features (Setting 3 of Experiment 3): Random Forests have been trained on $D_{M,k}$ for $M \leq 4$ with different time-contiguities $k$. Performance on $D_{4,k}$ has been evaluated by six times 10-fold spatial cross validation. Column (a) shows the average performance over all 60 station splits for three performance measures. Column (b) shows the average performance gain relative to the best case of $k = 1$, see Eq. (5) for the definition of performance gain. Errorbars illustrate the standard deviation. Column (c) shows the minimal performance gain. Across each row the same performance measure is considered.





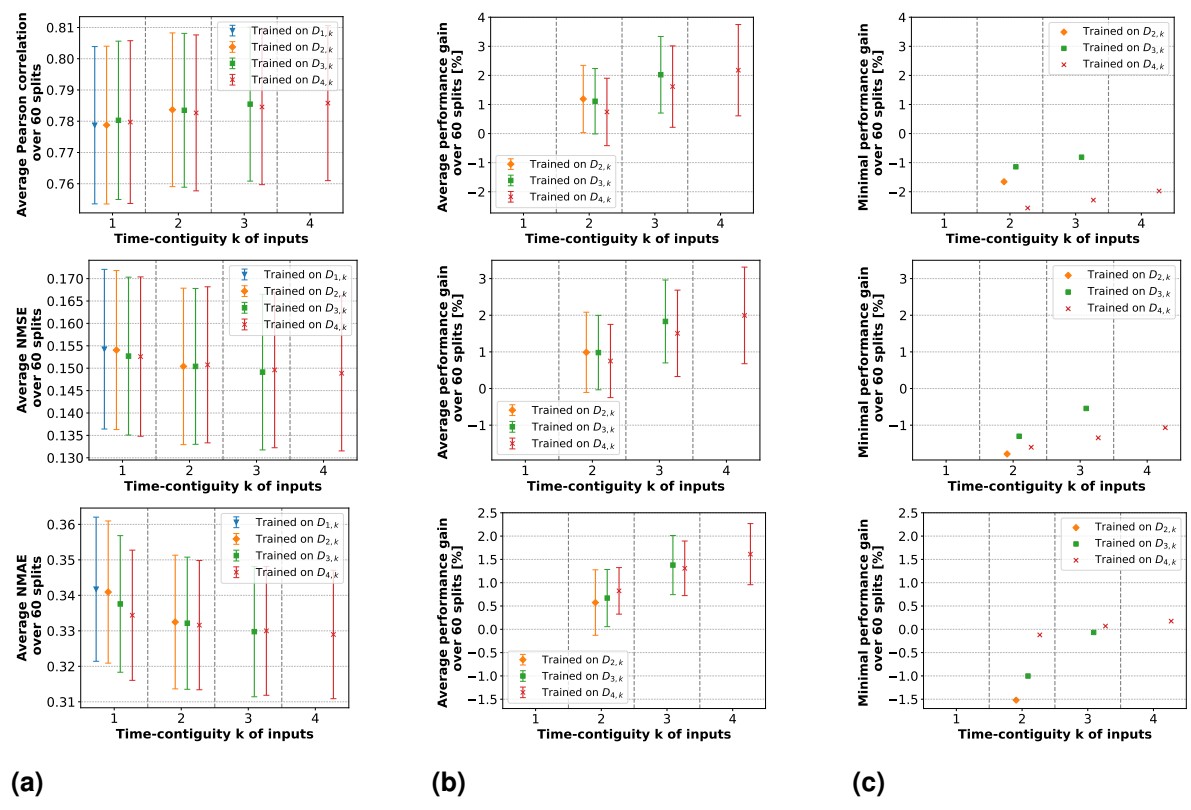

**Figure D2.** Same as Fig. D1, but tropospheric $NO_2$ VCDs were excluded from input features (Setting 2 of Experiment 3).





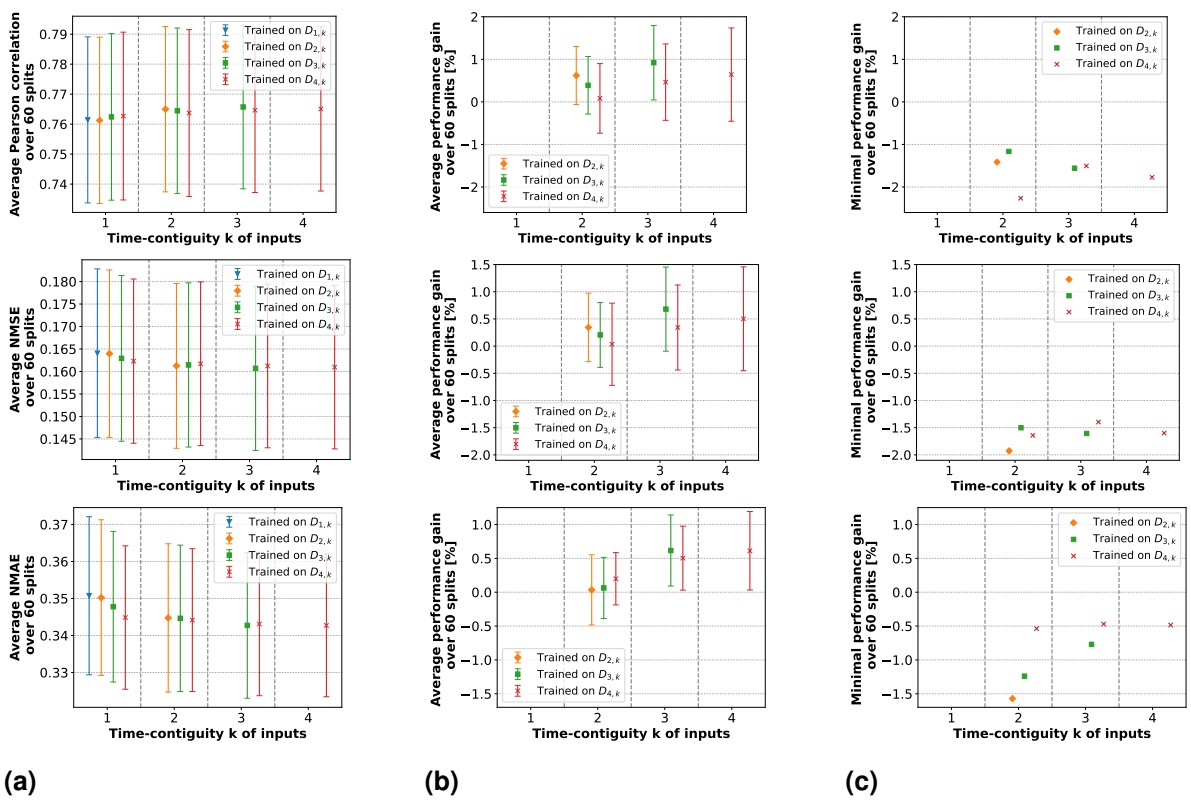

**Figure D3.** Same as Fig. D1, but tropospheric $NO_2$ VCDs, latitude and surface height were excluded from input features (Setting 4 of Experiment 3).





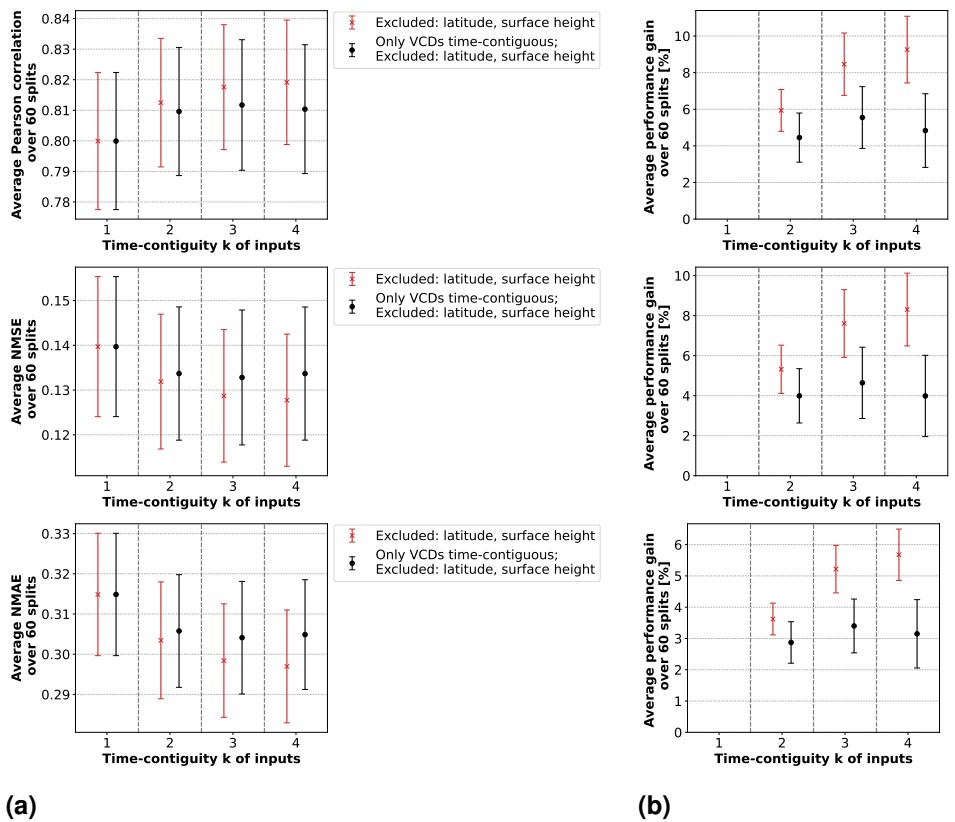

**(a)**                                        **(b)**

**Figure D4.** Random Forests: Selection of input features is as in Setting 3 of Experiment 3, i.e. latitude and surface height are excluded. Comparison of two cases: First, only time-contiguity of tropospheric $NO_2$ VCDs is exploited. Second, time-contiguity of all (time-dependent) input features is exploited, which is exactly Setting 3 of Experiment 3. Models have been trained and tested on $D_{4,k}$ for different time-contiguities $k$. Column (a) shows the average performance from six times 10-fold spatial cross validation and column (b) shows the average performance gain (Eq. (5)).

*Author contributions.* Janek Gödeke is the main author of this study, and planned and conducted the experiments. Andreas Richter and Kezia Lange provided GEMS data. Peter Maaß, Andreas Richter and Kezia Lange contributed to the design of the study and the discussion of results. Hyunkee Hong, Hanlim Lee and Junsung Park provided in-situ data and expertise on GEMS measurements. All authors contributed to the manuscript.

*Competing interests.* At least one of the (co-)authors is a member of the editorial board of Atmospheric Measurement Techniques.



*Acknowledgements.* We thank the National Institute of Environmental Research (NIER) of South Korea for providing GEMS lv1 data and financial support (NIER-2022-04-02-037). Hersbach et al. (2018) was downloaded from the Copernicus Climate Change Service (2023). The results contain modified Copernicus Climate Change Service information 2020. Neither the European Commission nor ECMWF is responsible for any use that may be made of the Copernicus information or data it contains. We thank the Korean Ministry Of Environment and NIER for providing the in situ measurements of surface $NO_2$. Janek Gödeke and Kezia Lange acknowledge funding by the Deutsches
Zentrum für Luft- und Raumfahrt (grant no. 50 EE 2204). Further, we thank Pascal Fernsel from the University of Bremen for fruitful discussions and feedback.



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
