# Peer review of "Hourly surface nitrogen dioxide retrieval from GEMS tropospheric vertical column densities: Benefit of using time-contiguous input features for machine learning models"

_EGUsphere, 2024_

## Author Comment (AC1)

**Reply to Referee 1 (Preprint egusphere-2024-3145 - Hourly surface nitrogen dioxide retrieval from GEMS tropospheric vertical column densities: Benefit of using time-contiguous input features for machine learning models)**

February 27, 2025

We would like to thank the Editor and the Referees for their comments and suggestions which helped us to improve the quality of our manuscript.

The following major changes were implemented in the revised version of the manuscript:

- A flowchart for the data processing workflow was added (see Fig. 2).

- A table with detailed information on the temporal and spatial data resolution and preprocessing steps was added (see Table 2).

- Some of the models in Section 5.3 were trained again and tested on seasonal data (see the new Section 5.4).

- These models were tested for different times of the day (see Section 5.4).

- We increased the size of symbols within plots for better readability.

Below, we give a point-by-point response to your review.

Janek Gödeke

**General Comments**

- (...) The figures and results clearly indicate the use of earlier data improves the performance of these machine learning models overall. I would have liked to see a small discussion about how these improvements might change over the course of a day. The GEMS observations are probably much less accurate in the morning and late evening (high angles, less sensitivity to the surface). The morning is furthermore limited in earlier time-contiguous observations but the evening is not. How does the performance of the final result change as a function of time? (...)

  Thank you for the suggestion! We have analyzed the test performance of some of our models from Section 5.3 (trained on whole-day data) for different times of the day and incorporated a corresponding discussion into the new Section 5.4. Additionally, we have mentioned in the outlook (Section 6) that a potential avenue for future research could involve training models specifically on subsets of data, such as morning-only data. It would be interesting to investigate whether this specialization might lead to further improvements in performance. However, implementing such an approach would necessitate tailored, time-specific hyperparameter tuning, which is beyond the scope of this study. Therefore, we only mention this as an outlook.

- Two rather basic models are used (linear regression and Random Forest), which the authors chose to more easily isolate the performance changes. It's not clear how the performance with time-contiguous data would change in other model setups. Do you expect those to have the same gains in performance?

  We agree that Random Forests are basic models in the sense of a limited number of tunable hyperparameters, but for regression tasks, they are in general powerful and competitive.

  At the outset of this study, we also experimented with Neural Networks (NNs) for estimating surface $NO_2$. While we observed similar results to those obtained with Random Forests, the training time for NNs was considerably longer. Therefore, and due to the large number of hyperparameters and architectural design choices for NNs, conducting as many experiments with NNs as we did with Random Forests would have been outside the scope of our study. This is why we chose to focus on Random Forests, but we expect similar performance gains also for Neural Networks.

  In the revised manuscript, we added a remark to the introduction of Section 4, in which we mention that we also did a few experiments with NNs and observed similar performances. But since they were much more time-consuming, we could not do the same number of experiments as we did for Random Forests.

**Specific Comments**

- Line 44: Change to "the measurement of lower tropospheric gases is not accurate"
  Thank you for the suggestion. We have changed that in the revised manuscript.

- Line 44: "This is why most studies estimated daily" doesn't follow from your previous statement. The estimate they give is still at a specific time, not a daily average which is implied here. Clarify this sentence.
  In fact, there exist both types of studies: Kim et al. (2017) predict surface $NO_2$ at the specific satellite observation time. Di et al. (2020) predict daily averages of surface $NO_2$. Therefore, we changed the sentence as follows: "Since satellites in low-earth orbits provide observations at most once a day, most studies either predicted surface $NO_2$ at this specific satellite observation time (e.g., Kim et al. (2017)), or they estimated daily (e.g., Di et al. (2020)), monthly or annual averages of surface $NO_2$."

- Line 105: I'm confused... where did j come from? The above equation uses t-k+1 (no t-j mentioned.)

  Thank you for your comment. In the input vector (line 106 in the revised manuscipt), $t - k + 1$ refers to the earliest time since $k$ is the time-contiguity. On the other hand, $j$ is a variable which takes values in the set $\{0, 1, ..., k - 1\}$. So $t - j$ stands for all times between $t - k + 1$ and $t$. We have specified this in line 107 of the revised manuscript.

- Line 148: Would be useful for context to summarize accuracy of the NO2 product you use, both for troposphere and stratosphere. And how does this change over a day?
  We agree that an investigation of the impact of uncertainties in the satellite columns on the predicted

surface concentrations would be interesting, but unfortunately, the GEMS IUP-UB product does not yet have full error propagation. The tropospheric NO2 VCD error is therefore estimated with 25%. The main uncertainty results from the assumptions used in the calculation of airmass factors, in particular for surface reflectivity, NO2 vertical profile and aerosol loading. As the reviewer points out, uncertainties are expected to be larger in the morning when the boundary layer is shallow and smaller around noon and in the evening. Uncertainties introduced by the stratospheric correction can be important over clean regions but can be neglected over pollution hotspots. We have added this information to Section 2.1.1 of the revised manuscript.

- Line 153: The TM5 model may leave residual structure in the results... maybe mention resolution here.

  The TM5 model has an hourly temporal resolution with a spatial resolution of $1° \times 1°$. As the model a priori is interpolated in space and time, no obvious structures from the coarse model resolution are visible in the data, but the lack of detail still may impact the results. We included this information to Section 2.1.1 of the revised manuscript.

- Line 175: What kind of sensors are used? What is accuracy of the sensors?

  Thank you for this question. We have been informed that the instruments utilize the chemiluminescence method, as described by Kley and McFarland (1980, Chemiluminescence detector for NO and NO2). We have included this information to Section 2.1.3 of the revised manuscript. However, we were also advised that the specific types of instruments may vary, along with their accuracy. We do unfortunately not have detailed information regarding these variations.

- Line 183: "We assume" – this seems like something that should be clear in a user guide or the information could come from the data producers upon request. Is this a fact or are you really making an assumption? Without more information it could also be assumed that 1:00UTC is describing the monthly average from 00:30-1:30 UTC. I generally find the time stamp discussion confusing. Wouldn't it make sense to label this example as 2021/01/23/02 since two datasets at least are occurring around 2:00 UTC?

  Unfortunately, there is no user guide available for the data. Within the dataset there is only one time label given per data point, e.g. 01:00 UTC. Therefore, we inspected the correlation between in situ surface $NO_2$ and VCDs at different hours, which suggested that the label 01:00 UTC in the in situ dataset refers to measurements between 01:00 and 02:00 UTC. However, in response to the reviewer's comment, we enquired again about the correct interpretation of the time label in the in-situ data and were informed that they indeed should be read as indicating measurements for the previous hour. This misinterpretation of the data is very unfortunate and could only be fixed by repeating all steps of the analysis, which is not possible at this point. However, the above mentioned tests showed only a small dependency towards changing the interpretation of the data by one hour, which gives us confidence that the conclusions of the manuscript are not affected by this mistake.

  Regarding the time stamps, we chose the same stamp that is used also within the VCD dataset. By that, we wanted to avoid confusion when using the VCD dataset.

- Line 209: Maybe I don't know enough about how these models work, but I don't understand how these negative values can be excluded, or why they have to be. Can you give some more justification? If the model is trained on a dataset that is biased at low column values of NO2, how does this affect results? If you don't care about the bias but can't handle negatives, why not add a background amount to make all the negatives positive to maximize use of all data? If you want to use the column values later to estimate surface NO2 in a given location but have negative values and haven't considered them in the model, how can these be used?

  Negative VCDs, so negative concentrations, have no physical meaning. This is why we excluded them from both the training and the test data to increase the quality of the dataset. By doing this consistently on both training and test data, the model does not suffer from any bias, because the model is trained and tested on the same type of data. One should not test these models on data points with negative VCDs (then we would agree that there could be some bias).

  For future tasks, one could also train and test new models on the larger datasets in which negative VCDs are included. However, we doubt that increasing the dataset is always beneficial, as it could also be disadvantageous, because it could make the data less interpretable by the model.

- Table 1: I think it would be useful to re-define N and give its unit here in caption.

We have added this information to the revised manuscript.

- Line 314: I'm not really clear about why latitude should get included at all as a feature in the first place. It's good to see later that its inclusion doesn't matter much, as the tropospheric VCD should have very little dependence on latitude in a physical way. Presumably the correlation in Table B1 is moderately high because in Korea the NO2 sources are dominated by a few cities including Seoul in the North, but the latitude is not the cause of enhanced tropospheric NO2. It could be important for other gases and larger domains, but not trop NO2 in a tiny area like South Korea.

  We agree that the inclusion of the coordinates might be problematic and shouldn't matter much. However, other studies have used spatial coordinates for predicting surface $NO_2$. Mainly over large regions, such as the USA (e.g., Gharemanloo et al. (2021)) or China (e.g., Li et al. (2022), Qin et al. (2020)). But also over smaller regions, such as over Switzerland (e.g., deHoogh et al. (2019)).

  We took spatial coordinates (longitude/latitude) into consideration during feature selection (Section 3.1) because we wanted to check:

  - Although spatial coordinates only slightly differ within Korea, couldn't there be a small helpful information for the model from spatial information?
  - Is there an additional risk for spatial overfitting when taking spatial coordinates as an input? This is why in Experiment 3 (Section 5.3) we made the same analysis without using latitude as an input.

- Figures 2 and 3: Not a big deal but I'm not sure why left column has to be included... seems redundant with middle column which provides a more complete result.

  We agree that the left column is a bit redundant. Our motivation for including it, though, is to stress that although the standard deviations in the middle column are quite large, the curves of individual station splits are more or less parallel. We think this will make it easier for the reader to understand that there is always a benefit from time-contiguous inputs, for every individual split and not only on average.

- Line 653: Here and earlier, I'm not clear why you would want to use this model outside of Korea with no VCD input (also, the focus of the paper seems to be GEMS – i.e., satellite observations). Can you elaborate under what circumstances this would be useful? I would expect it to be pretty inaccurate without the VCD, especially in regions with no monitors, and not as useful as a physical model output from something like CAMS or GEOS-CF.

  Excluding VCDs as an input for the models did lead to worse predictions over Korea (see Section 5.3). However, we would actually have expected an even larger decay in performance. Nevertheless, we believe that VCDs as an input will prevent the model from spatial overfitting, which may be a less significant effect for a small area like Korea. This motivates the investigation within future work whether the models without VCD-input will indeed perform much worse at locations outside of Korea. Further, a model that also works outside the region at which it has been trained is desirable in practice, because by that one could get a broad picture about surface $NO_2$ pollution, even if no data from ground monitors are available.

**Technical Comments**

- English issues

  We have corrected the English issues that you have pointed out.

- Figure 2 and 3: In these and other figures, the linewidth, symbol size and sometimes font size are very small and hard to read on my screen. There are not many points, so there is a lot of room to improve the figures by making lines and symbols larger in future plots.

  Thank you for this suggestion. We have used larger symbols and lines in the updated version of the manuscript.

---

## Author Comment (AC2)

**Reply to Referee 2 (Preprint egusphere-2024-3145 - Hourly surface nitrogen dioxide retrieval from GEMS tropospheric vertical column densities: Benefit of using time-contiguous input features for machine learning models)**

February 27, 2025

We would like to thank the Editor and the Referees for their comments and suggestions which helped us to improve the quality of our manuscript.

The following major changes were implemented in the revised version of the manuscript:

- A flowchart for the data processing workflow was added (see Fig. 2).

- A table with detailed information on the temporal and spatial data resolution and preprocessing steps was added (see Table 2).

- Some of the models in Section 5.3 were trained again and tested on seasonal data (see the new Section 5.4).

- These models were tested for different times of the day (see Section 5.4).

- We increased the size of symbols within plots for better readability.

Below, we give a point-by-point response to your review.

Janek Gödeke

**General Comments**

**Data Processing**

The data processing methodology is unclear and lacks sufficient detail. The authors should enhance this section by:

- Including a flowchart in the Data section to visually illustrate the entire data processing workflow

  Thanks for the suggestion, we have added a flowchart to Section 2 that visualizes all data processing steps (see Fig. 2).

- Providing a table detailing each data source, including the spatial and temporal resolution, and any preprocessing steps applied to the input datasets.

  Information about preprocessing, as well as the spatial and temporal resolution of the data, is provided in Sections 2.1 and 2.2. However, we agree that a compact overview in the form of a table enhances clarity. We have included a table into the revised manuscript (see Table 2).

**Satellite and Ground Station Pairing**

- The statement "we associated the location of an in situ station with the VCD pixel or meteorological pixel whose center is nearest to the station's location" needs clarification. Does this refer to the center of the satellite pixel?

  Yes, it refers to the center of the satellite pixel since we do not apply a regridding. Thank you for this remark. We have clarified this in line 192 of the revised manuscript.

- If multiple ground stations fall within the same satellite pixel, how are these handled? Are they averaged, or is one selected?

  If multiple stations fall within the same satellite pixel, data points are generated separately for each station and included in the dataset. We are aware that if these stations measure different values, this adds noise to the dataset. However, this has the advantage of reducing the risk that the model overfits to the training data.

- GEMS pixel locations vary slightly with each scan due to orbital and observation geometry. Did the authors regrid the satellite data before co-location to ensure consistency?

  No, we did not regrid the satellite/VCD data onto a universal grid. This decision was made to minimize data manipulation and preserve the integrity of the original measurements.

**Characteristics of Ground Stations**

- What type of instruments are used at the ground stations? For example, are they chemiluminescent analyzers?

  Thank you for your question. We have been informed that the instruments utilize the chemiluminescence method, as described by Kley and McFarland (1980, Chemiluminescence detector for NO and $NO_2$). We have included this information to Section 2.1.3 of the revised manuscript. However, we were also advised that the specific types of instruments may vary, along with their accuracy. We do unfortunately not have detailed information regarding these variations.

- Ground stations are often categorized as urban, background, or roadside. Did the authors use all station types, or restrict their analysis to specific types? The representativeness of the training data depends on this choice.

  No stations were excluded based on their location. As shown in Figure 1, stations are distributed across both urban and rural areas. As the spatial resolution of the satellite data is limited and vertical profiles vary between location types, it is to be expected that the relationship between $NO_2$ column and surface concentration is different for different location types. However, the approach taken here was to use all stations in the same way, and not to limit the training dataset to a certain type of station or to provide the station type as additional parameter.

**Temporal Input and Data Loss**

- The model will not produce predictions for the first few hours of each day, creating data gaps.

  Yes, if the model receives time-contiguous inputs, it is unable to make predictions for the initial hours of each day. This is a limitation which we included in the revised manuscript, see line 272 in Section 2.3.

- Cloud cover and other issues affecting satellite measurements in prior hours can propagate errors into the current hour's input, resulting in significant data losses during training and prediction. This cascading loss reduces the dataset from over 1.3 million data points for a 1-hour input window to approximately 350,000 for a 5-hour window. The authors should provide a clear justification for accepting this trade-off between increased data gaps and potential gains in model accuracy.

  To address this trade-off between data gaps and improved model accuracy, we have designed Experiment 2 (see lines 330–339 in the revised manuscript). Our findings indicate that models trained on the largest dataset ($\geq 1$ million points) without time-contiguity perform worse than models trained on smaller datasets with time-contiguity. These results are discussed in detail in Section 5.2.

- Additionally, the authors should evaluate and discuss how these data gaps impact not only the training and validation phases but also the model's predictions and its applicability to real-world scenarios. This includes addressing potential limitations in the model's ability to generalize when encountering similar conditions in operational or extended applications.

  If a model is trained with time-contiguity, one cannot apply it to data points for which the time-contiguous features do not exist. Our study should be understood as follows: If time-contiguous features are not available for some data point, then use a model that has been trained without time-contiguity on the large dataset. However, as soon as we want to make a good prediction for a data point for which time-contiguous features are available, we have shown in Experiment 2 that better predictions are made by time-contiguous models. In real-world scenarios, this means that time-contiguous models support the non-time-contiguous model whenever this is possible. In the revised manuscript, we stressed this point in an additional remark after the rule of thumb, see lines 713-715, or already in line 594-596.

**Justification for Input Variables and Preprocessing**

- The paper lacks justification for the selection of input variables. A sensitivity analysis or variance inflation factor (VIF) analysis should be conducted to ensure the chosen variables are non-redundant and significant

  In Section 3.1, we describe the feature selection process used in our study. We used a simple criterion: We computed the correlation between VCDs and potential input features on the training datasets. Then we selected those features whose correlation to VCDs is at least 0.1, so that the features have a better chance of being significant. We cross-validated the correlation on 200 different potential training sets and observed that this criterion is very stable, meaning that (almost) always the same features are selected according to this criterion. However, we do not put a focus on redundancies among the features because of the following:

  Using time-contiguous inputs will presumably introduce some redundancy to the inputs if some of the features do not change much over time. A complete sensitivity analysis would not only require examining the choice of features, but also assessing the optimal time-contiguity for each feature. A complete sensitivity analysis would be out of scope for our study, and we chose not to deviate from our main focus, which is evaluating whether time-contiguous inputs are beneficial.

  As a compromise, we did a sensitivity analysis in Experiment 3 regarding three input features: VCDs, latitude and surface height. We explain the rationale for inspecting these features in Section 3.2 (lines 340–348 in the revised manuscript), and the results are discussed in Section 5.3.

- The input variables differ in units and magnitudes, which could cause instability in model performance. Did the authors scale, normalize, or log-transform these variables before training? This critical preprocessing step is missing from the discussion.

  Thank you for this question. Yes, we applied an affine linear transformation to each feature, such that the mean of the training data is 0 and the standard deviation is 1. We have added this information to the end of Section 2.3 of the revised manuscript.

**Choice of Models**

- The authors used Random Forest and linear regression but did not justify these choices.

  The motivation for using Random Forests and linear regression is given in the introduction of Section 4: Random Forests have only few hyper parameters to tune. Therefore, one gets a clearer insight into whether time-contiguous inputs are beneficial. Additionally, Random Forests are well-suited and powerful for regression tasks. Although linear regression models are not competitive, they give a first insight into the experiments.

- More advanced machine learning methods, such as Artificial Neural Networks (ANN), Recurrent Neural Networks (RNN), or Convolutional Neural Networks (CNN), have been shown to better handle non-linear relationships and spatio-temporal dependencies in atmospheric data. The authors should explain why these advanced methods were not used or compare their results to them.

  At the outset of this study, we also experimented with Neural Networks (NNs) for estimating surface $NO_2$. While we observed similar results to those obtained with Random Forests, the training time for NNs was considerably longer. Therefore, and due to the large number of hyperparameters and architectural design choices for NNs, conducting as many experiments with NNs as we did with Random Forests would have been outside the scope of our study. This is why we chose to focus on Random Forests, but we expect similar performance gains also for Neural Networks.

  In the revised manuscript, we added a remark to the introduction of Section 4, in which we mention that we also did a few experiments with NNs and observed similar performances. But since they were much more time-consuming, we could not do the same number of experiments as we did for Random Forests.

  We have avoided comparing our results, obtained with Random Forests, to methods from other studies due to the use of different datasets, primarily from different regions of the world. Such comparisons could risk being unfair in either direction.

**Handling of Negative Values**

- The authors ignored negative GEMS VCD values, which will bias the average toward positive values. Justification is needed for this choice.

  Thank you for the comment. However, we do not think the models suffer from any bias due to the decision of excluding negative VCDs, because the models are both trained and tested on this type of data. One should not test these models on data points with negative VCDs (then we would agree that there is some bias).

  Negative VCDs, so negative concentrations, have no physical meaning. This is why we excluded them from both the training and the test data to increase the quality of the dataset. Excluding negative input VCD values from a training set does not create a low bias in the target quantity unless you later feed negative VCD values to the model.

- Similarly, were there negative values in the in-situ measurements? If so, how were these handled? This needs to be explicitly discussed.

  Thanks for this question. No, there were no negative values in the in-situ measurements. We have added this information to the end of Section 2.2 (lines 224-226) and to the description of the data-preprocessing in Section 2.3.

**QA Value Threshold and Bias**

- The authors only used data with QA values equal to 1. This choice filters out cloudy conditions but potentially introduces a clear-sky bias since cloudy conditions can be associated with higher aerosol or NO2 levels. The authors should address this limitation and quantify its impact on results.

  The decision to only consider QA values equal to 1 implies that the model is not (reliably) applicable to situations in which there are cloudy conditions. We have added a note on that at the end of Section 2.2. Additionally, we have remarked that a potential future direction of this work could involve examining the effects of lowering the QA threshold. This would result in a larger, but more complex, dataset.

**Inclusion of Latitude**

- Including latitude as an input variable needs further justification, as the latitudinal variation over South Korea is minimal. The authors should explain the rationale behind this decision.

  We agree that the inclusion of the coordinates might be problematic. However, other studies have used spatial coordinates for predicting surface $NO_2$. Mainly over large regions, such as the USA (e.g., Gharemanloo et al. (2021)) or China (e.g., Li et al. (2022), Qin et al. (2020)). But also over smaller regions, such as over Switzerland (e.g., deHoogh et al. (2019)).

  We took spatial coordinates (longitude/latitude) into consideration during feature selection (Section 3.1) because we wanted to check:

  - Although spatial coordinates only slightly differ within Korea, couldn't there be a small helpful information for the model from spatial information?
  - Is there an additional risk for spatial overfitting when taking spatial coordinates as an input? This is why in Experiment 3 (Section 5.3) we made the same analysis without using latitude as an input.

**Section-Specific Comments**

**Section 5.2**

The atmospheric lifetime of NO2 varies with season and time of day, and this variability likely influences model sensitivity. The authors should:

- Conduct and present seasonal and diurnal sensitivity analyses to account for these variations.

  Thank you for the suggestion. We have trained some of the models discussed in Section 5.3 and evaluated their performance on seasonal datasets. We included this to a new Section 5.4 in the revised version of the manuscript. Additionally, we examined their performance over the course of the day.

- Address potential biases from the limited temporal scope of training data (January 2021 to November 2022). For instance, why was data from December underrepresented, and why were only 23 months used instead of two full years?

  We received the in situ dataset at the beginning of December 2022. Therefore, data from December was not available at that time. We made a note on that in Section 2.1.3 of the revised manuscript. Further, a short remark of a potential bias due to the Covid-19 pandemic, has been added to the end of Section 6 of the revised manuscript.

- Discuss whether differences in valid data points across seasons (e.g., more data in summer due to fewer clouds) lead to seasonal biases in model training.

  In fact, after applying a filter for the qa-value, there are less data points in summer available, due to the monsoon. We added Table 3 to the revised document, which shows the contribution of each season to the total number of data points. Further, we have inspected the performance of some models from Section 5.3 at all different seasons. Details were added to the new Section 5.4.

**Section 5.3**

- The prediction maps show that the model has been applied beyond South Korea, including regions over the ocean, Japan, and North Korea. The authors should: Validate the model's performance in these regions by comparing predictions to in-situ measurements from other countries, such as Japan. This would demonstrate the model's transferability across different geographies.

  Thanks for the suggestion. Evaluating the model's performance outside of Korea would be an interesting extension of our work. However, accessing datasets from regions such as Japan and studying the model's performance there is left as a future task. This is mentioned in Section 6.

- The prediction maps also exhibit noticeable grid structures, likely originating from the meteorological ERA5 dataset. Did the authors interpolate the ERA5 data to reduce these artifacts? If not, why?

  We did not interpolate the ERA5 data to reduce these grid artifacts, as we did not want to modify the data solely for the purpose of improving the appearance of the plots. Our focus was on preserving

the integrity of the original dataset, and we believe that these artifacts are a natural characteristic of the data.

- Clarify how gaps in GEMS data (e.g., due to cloud cover) were handled during prediction. The maps show no missing areas (Figure 9 and 10), suggesting the model was applied to cloudy data despite such data being excluded during training. Discuss the implications of using potentially contaminated data and its impact on model accuracy.

    The maps do show missing areas, because gaps in GEMS data were not considered during the prediction. Figures 9 and 10 (in the revised manuscript, these are Figs. 10 and 11, respectively) display these missing areas as black pixels. This is also noted in the description of Figure 9. In the caption of Figure 9, we state: "The black mask indicates missing data, e.g. due to clouds."

---

## Author Response (AR2)

**Reply to both Referees (Preprint egusphere-2024-3145 - Hourly surface nitrogen dioxide retrieval from GEMS tropospheric vertical column densities: Benefit of using time-contiguous input features for machine learning models)**

April 9, 2025

We would like to thank the Editor and the Referees for their comments and suggestions which helped us to improve the quality of our manuscript.

The following major changes were implemented in the revised version of the manuscript:

- In Figures 10 and 11, we have changed the coastlines to white and removed the black frame around the colorbars. This ensures that there is no ambiguity regarding the black mask for missing data.

- We clarified the choice of latitude as an input feature in lines 344-353.

- We remarked in lines 223-226 that experiments on a small dataset indicated that filtering negative VCDs has presumably a very small impact on the performance and would be neglected for future studies.

- In lines 603-607 of Section 5.2, we mentioned an interesting task for future work: Inspecting further potential differences when switching between models of different time-contiguity.

Below, we give a point-by-point response to your reviews.

Janek Gödeke

**Reply to Referee 1**

- I am not convinced with the reply about the exclusion of negative values. I note that the other reviewer also had a similar concern in their original review. In their response, the authors note that "Negative VCDs, so negative concentrations, have no physical meaning. This is why we excluded them from both the training and the test data to increase the quality of the dataset." It is not true that negative values have no physical meaning in these kind of satellite data, which are in effect differential measurements. There are a number of reasons that negative values many occur. The first is fitting a spectrum with random noise. Consider a measurement of a completely "clean" atmosphere with no tropospheric NO2 and a column of zero molecules/cm2. If this is measured by a detector with any noise, we would expect the observations to be distributed about zero (i.e., half will be positive and half will be negative.). Excluding all negative values in an analysis with this data will bias results high, where in truth each small negative value is more or less as significant as each equally small positive value.

  In addition, systematic uncertainties in spectral fitting inputs, like cross sections, could cause negative biases in the data in clean regions. Furthermore, the fact that a tropospheric column is being derived from a total column measurement using an estimated stratospheric column is another source of a potential negative bias in a tropospheric column. Even if the model cannot deal with negative values (which I guess is the case), I think there needs to be more of a discussion of how excluding these values could affect the results, rather than brushing over this point. I would think as well as affecting model performance, it needs to be explained what will happen to the negative VCDs when the final model is applied to a map of VCDs to derive concentrations. Can these be used, or are you at risk of losing data over clean regions that are actually useful (again, potentially biasing results)?

  We agree that negative columns must not be removed when computing averages of satellite data as that would create a high bias. Here, the situation is slightly different as not the target quantity (the in-situ observations) but one of the inputs (the satellite data) is filtered. In our opinion, there is no reason to expect a high bias of the predicted surface concentrations just because one of the input variables to the random forest is limited to positive values.

  However, we agree with the reviewers that negative satellite columns are usually found over regions with low tropospheric columns, and therefore, the filter we applied leads to a loss of predictions for such situations. In this sense, the ensemble of predicted values may be biased high.

  We tested the effect of the filter on a small data set and found very little changes. This is probably due to the fact that applying this filter only leads to a reduction of the dataset by less than 0.5%. In hindsight, it probably was not useful to implement this filter. Unfortunately, repeating the study without this filter would be a large effort at this point, but this lesson will be taken into account for future work. In the revised manuscript, we mentioned this in lines 223-226.

  When Random Forests are trained on non-negative VCDs only, they are still able to make reasonable, but potentially biased, predictions over clean regions with negative VCDs as inputs. In this case the Random Forests would treat negative VCDs as being zero.

- I am also still confused by the use of latitude as a predictor variable in such a small region as Korea (even if the justification is that it has been used in a previous paper over Switzerland). I am still not convinced it is a useful variable on which to focus, but I suppose at this point would it would require a lot of work to redo the model and paper. The authors responded to me and the other reviewer (also with this concern) but I am not sure they have changed the paper in any way to address this comment. It might be helpful to add a line or two to the paper to clarify the choice of this predictor.

  Thank you for pointing out that we had not clarified the choice of latitude in the manuscript. In Section 3.2, we have now added in lines 344-353 some clarification for the selection of latitude as a predictor, in accordance with our response to the first review. Regarding the consideration of input features without latitude (but with surface height), as well as an alternative choice for feature studies in Experiment 3, implementing these changes would indeed require extensive model training and tuning, necessitating substantial revisions to the paper.

**Reply to Referee 2**

- Your response mentions the use of "models" and a "rule of thumb" while utilizing an ensemble of models trained on different time-contiguous features. However, critical questions regarding the model's practical application and potential biases remain. While you suggest using a Random Forest trained with k = j' + 1 when time-contiguous features are available, and switching to a model trained without time-contiguity when they are not. Did the authors analysis the bias or any potential differences when switching among these models trained with different time-contiguous features?

  In Experiment 2, we analyzed the difference between all different models in terms of their prediction accuracy with respect to different loss functions. For all five considered loss functions, we made the same observation that the usage of time-contiguous models are beneficial and similar rule of thumbs apply, see Table B2. However, we did not systematically assess other potential differences that may arise when switching between models trained with different time-contiguous features. For instance, it remains an interesting task for future studies to investigate whether the ensemble of such models yields consistent combined spatial patterns in predicted surface $NO_2$. We included this as a remark in lines 603-607.

- In your response, you state, 'Negative VCDs, so negative concentrations, have no physical meaning. This is why we excluded them from both the training and the test data to increase the quality of the dataset.' While it is true that negative concentrations are not physically realistic in the absolute sense, negative measured values can and do occur in real-world datasets due to measurement noise, particularly when the true values are close to zero. These negative values are not necessarily indicative of data quality issues but rather a reflection of the inherent uncertainty in the measurements. Furthermore, your claim that excluding these negative values does not introduce bias is actually incorrect. By systematically removing negative values, you are artificially shifting the mean of the dataset towards positive values. This will inevitably introduce a positive bias in any model trained on this altered dataset, regardless of whether the test data also lacks negative values. While your model might perform well on your artificially positive-shifted test data, it will not accurately reflect real-world scenarios where extremely low values are sometimes measured as negative. Skipping that part will definitely result a positive bias, especially for regions with low values. Also, I cannot see why the authors have to apply this additional filtering and cause addition missing data in their prediction. Therefore, the argument that the model will only be used on data without negative values, as you imply, is not well supported.

  We agree that negative columns must not be removed when computing averages of satellite data as that would create a high bias. Here, the situation is slightly different as not the target quantity (the in-situ observations) but one of the inputs (the satellite data) is filtered. In our opinion, there is no reason to expect a high bias of the predicted surface concentrations just because one of the input variables to the random forest is limited to positive values.

  However, we agree with the reviewers that negative satellite columns are usually found over regions with low tropospheric columns, and therefore, the filter we applied leads to a loss of predictions for such situations. In this sense, the ensemble of predicted values may be biased high.

  We tested the effect of the filter on a small data set and found very little changes. This is probably due to the fact that applying this filter only leads to a reduction of the dataset by less than 0.5%. In retrospect, it probably was not useful to implement this filter. Unfortunately, repeating the study without this filter would be a large effort at this point, but this lesson will be taken into account for future work. In the revised manuscript, we mentioned this in lines 223-226.

- Thank you for pointing out the black mask indicates missing data in your figures. But I would suggest using a different color for missing values, as your coast lines and lowest value of your colorbar are also black.

  Thank you for the suggestion! We experimented with different colors for masking the missing data but were not satisfied with the results. For example, a white mask drew too much attention away from the rest of the image. Therefore, we opted for the following alternative: We retained the black mask for missing data but changed the coastline color to white. Additionally, we removed the black frame around the colorbars to make it clearer that black is not the lower end of the color scale.